# Computational engineering of the polyester hydrolase PHL7 for efficient poly(ethylene terephthalate) degradation in biocatalytic recycling processes

Paula Blázquez-Sánchez [1,7], Jonas Gunkel[1,7], Abibe Useini [1,2,7], Alexander Zlobin[1], Jonathan D. Zakary[1], Andrea Schöler[1], Norbert Graefe[1,3], Felipe Engelberger [1], Filipa Cantanhede[1], Ronny Frank[4], Ziyue Zhao[5], Afsaneh Zarei[5], Erik Butenschön[5], Jörg Matysik [5], Wolfgang Zimmermann [5], Norbert Sträter[2], Christian Sonnendecker [5] ✉ & Georg Künze [1,3,6] ✉

Polyethylene terephthalate (PET) plastic waste causes serious environmental pollution due to insufficient recycling rates. Enzymatic PET depolymerization offers a sustainable recycling strategy, but limited stability and activity of current PET-degrading enzymes restrict practical implementation. Here, we engineer Polyester Hydrolase Leipzig 7 (PHL7), a PET hydrolase from a compost metagenome, to enhance its stability and catalytic performance under recycling-relevant conditions. Using Rosetta PROSS-based computational design combined with rational mutagenesis, we introduce up to 24 mutations, generating variants with melting temperatures of 88-95 °C and over 110-fold higher activity in 0.1 M phosphate buffer compared to the parent enzyme. Benchmarking shows that the best variants (R4M6, R4M9, and R4M10) match or exceed the performance of established engineered PET hydrolases, including ICCG and LCC-A2, and approach that of TurboPETase across multiple conditions. Under high substrate loadings, the PHL7-R4 variants degrade 75-78% of 10% (w/w) PET within 24 h at 65 °C, outperforming ICCG, while an optimized variant R4M10-H185Y achieves up to 84% degradation of 20% (w/w) PET. X-ray structure determination and molecular dynamics simulations reveal key stabilizing and activity enhancing mechanisms. These engineered PHL7 variants represent robust biocatalysts for scalable enzymatic PET recycling.

Polyethylene terephthalate (PET) is a synthetic polymer extensively utilized across various industries, notably in packaging and textiles, due to its robustness and versatility. The inadequacy of end-of-life management for PET, where a mere 23% of plastic waste has been recycled in 2020[1], underscores the critical need for innovative and sustainable approaches to mitigate the impact of plastic pollution[2]. Biocatalytic recycling of PET, leveraging the specificity and efficiency

of hydrolases, offers a promising avenue towards addressing the challenges of PET waste. PET hydrolases are capable of breaking down PET into its constituent monomers, terephthalic acid (TPA) and ethylene glycol (EG), opening the door to biochemical recycling[3–5].

Enzymatic PET degradation has been reported to be most effective at around the glass transition temperature of PET (ca. 65 °C), requiring highly thermostable PET hydrolases[6]. Thermostable PET

---

hydrolases can retain their catalytic activity at elevated temperatures throughout long reaction times, making them well-suited for industrial applications[7]. However, at high temperatures, recrystallization and physical aging of PET can lower the extent of PET degradation during longer reaction times. Thus, both high thermostability and high enzymatic activity are required to achieve high turnover rates under these conditions[7]. Furthermore, to improve enzyme stability and keep the pH level of the reaction constant, high-salt buffers are often used in lab-scale experiments. However, this should be avoided in industrial settings to reduce operational costs, ensure reactor longevity, and simplify downstream processing of the hydrolysis products[8]. While some PET hydrolases already perform well under low-salt conditions, others depend on elevated ionic strength to remain stable and catalytically active, rendering the mitigation of salt dependence a design goal for specific enzymes.

Considerable progress has been made in the development of effective PET hydrolases by harnessing enzymes from natural sources and optimizing their properties through iterative rounds of protein engineering[4,9]. Enzyme stabilization could be achieved by the introduction of disulfide and salt bridges at positions predicted by sequence- and structure-based analyses[10–14]. Furthermore, directed evolution and machine learning approaches have been applied to identify mutations that tailor enzymes to desired temperature, pH, or PET crystallinity ranges[15–18]. These efforts have culminated in the development of high-performance enzymes, such as engineered variants of *Ideonella sakaiensis* PETase (*Is*PETase)[11,15,16,19,20] and leaf-branch compost cutinase (LCC)[12,18,21–23], capable of efficiently degrading low-crystalline PET at elevated temperatures. For example, the optimized LCC variant ICCG is one of the most efficient PET hydrolases reported to date[12]. Other recent successes in engineering PET hydrolases include, among others, the enzyme variants LCC-LANL[18], LCC-A2[21], TurboPETase[24], and Kubu-P^MI2[25].

In this study, we aimed to develop a high-performance PET hydrolase by redesigning the polyester hydrolase Leipzig 7 (PHL7), a recently discovered hydrolase isolated from a plant compost metagenome[26]. PHL7 rapidly degrades low-crystalline PET at 70 °C and, therefore, is a good candidate for the development of biocatalytic PET recycling processes. However, the activity of PHL7 at high temperatures depends on high buffer concentrations. Whereas PHL7 maintains high activity for more than 24 h at 70 °C in 1 M phosphate buffer, it completely loses its activity in 0.1 M phosphate buffer within 24 h[26], indicating that the protein becomes unstable under these conditions. Thus, increasing the thermodynamic stability of PHL7, while also enhancing its catalytic efficiency, will be crucial for its deployment in industrial processes.

Building on insights derived from the PHL7 structure and guided by mutational hotspots identified in homologous PET hydrolases, previous design efforts applied targeted mutagenesis in PHL7 to boost its stability and activity[27,28]. Recently, further enhanced PHL7 variants were developed using directed evolution and computer-aided design strategies[29,30]. Notably, the designed variants PHL7-Jemez[29] and FlashPETase[30] demonstrated over two-fold higher PET degradation rates relative to PHL7 under their optimal reaction conditions in 1 M phosphate buffer. However, protein stability was only marginally increased for FlashPETase and even decreased for PHL7-Jemez, consistent with the low activity of PHL7-Jemez in 0.1 M phosphate buffer, equaling the activity of the wild-type (WT) PHL7. These observations emphasize that further improvements of PHL7 are needed to tailor it to industrial process conditions.

Accordingly, in this study, we have engineered PHL7 using a combined computational and rational design strategy to overcome its limited stability, catalytic efficiency, and strong salt dependence. This work has yielded multi-mutation variants with enhanced thermostability and PET degradation activity under low-salt and high-substrate-loading conditions, extending beyond the incremental improvements achieved in previous PHL7 engineering studies. Benchmarking across multiple reaction conditions shows that these PHL7 variants match or exceed the performance of recently reported high-performance PET hydrolases, establishing them as suitable biocatalysts for enzymatic PET recycling applications.

## Results

PHL7 was engineered through an iterative design workflow combining Rosetta-based protein modeling with rational mutagenesis to systematically enhance enzyme stability and activity (Fig. 1). Efficient exploration of a broad sequence and structural space by the PROSS algorithm[31] enabled the identification of beneficial multi-mutation combinations that would likely remain undiscovered in conventional single-site mutagenesis strategies. Successive design-test cycles across four engineering rounds, described below, identified mutations that optimized the balance between thermostability and catalytic efficiency. This approach established a robust framework for the rational improvement of PET hydrolases.

### Rational design of PHL7 identifies activity enhancing mutations

In previous research[27], we developed 16 single-point mutants of PHL7 by altering residues in its active site to increase its enzymatic activity and thermostability. The mutants were designed in a rational way by comparing the sequences and structures of PHL7 and homologous enzymes (i.e., LCC[32], ICCG[12], *Tf*Cut2[33], and *Is*PETase)[34,35]. We started the present study by combining those mutations (L93F, Q95Y, Q95G, L210T, and D233K) (Fig. 2b), and adding a new mutation inspired by FAST-PETase (Q175E)[16], to create a salt bridge with the neighboring R205. In FAST-PETase, this salt bridge is formed by the homologous residues E204 and K233.

We created 13 variants with two to five combined mutations (L93F/Q95G, L93F/Q95Y, L93F/L210T, Q95G/L210T, Q95Y/L210T, L210T/Q175E, L93F/Q95G/L210T, L93F/Q95Y/L210T, Q95G/L210T/D233K, L210T/Q175E/D233K, L93F/Q95G/L210T/D233K, L93F/Q95Y/Q175E/D233K, L93F/Q95G/L210T/Q175E/D233K) and evaluated their PET-hydrolytic activity on amorphous PET films at temperatures of 65 and 70 °C and at phosphate buffer concentrations of 1.0 and 0.1 M.

The results revealed a group of variants with significantly enhanced thermostability and reduced activity (Fig. 2a, red cluster), which includes Q95G/L210T, L93F/Q95G/L210T, Q95G/L210T/D233K, and L93F/Q95G/L210T/D233K. These variants exhibited melting temperatures of 82 °C to almost 84 °C, but their activity levels were reduced to 0.81–0.96-fold relative to the WT PHL7 (corresponding to PET weight losses of 66.2–78.7% after 14 h). All these variants include the Q95G mutation, which, as a single variant, demonstrated reduced activity but the largest melting temperature ($T_m$ = 81.1 °C) among the tested single-point mutations.

Another group of variants combined high thermostability and WT-like PET-hydrolytic activities (Fig. 2a, blue cluster). The variants L93F/Q95G/L210T/Q175E/D233K and L93F/Q95Y/Q175E/D233K showed $T_m$ values around 84 °C or higher. Q95G/L210T can also be included in this group, but its average activity was slightly below (0.96-fold) the WT level. Another set of variants, including L93F/Q95Y and L93F/Q95G, showed a moderate increase in thermostability ($T_m$ of 81.3 and 80.2 °C) and WT-like activity levels (Fig. 2a, yellow cluster). The most promising group of variants, demonstrating increased PET degradation activity as well as moderately increased thermostability (Fig. 2a, green cluster), includes L93F/L210T, L210T/Q175E/D233K, L210T/Q175E, Q95Y/L210T, and L93F/Q95Y/L210T, showing activity levels of 1.15-fold to 1.3-fold and $T_m$ values of 81.0–81.3 °C.

Noteworthy, Q95Y revealed a complex influence in multi-point variants. While the single mutation boosts activity, the activity of Q95Y/L210T is lowered compared to the L210T single mutation. Similarly, the activity of L93F/Q95Y/L210T is reduced compared to the

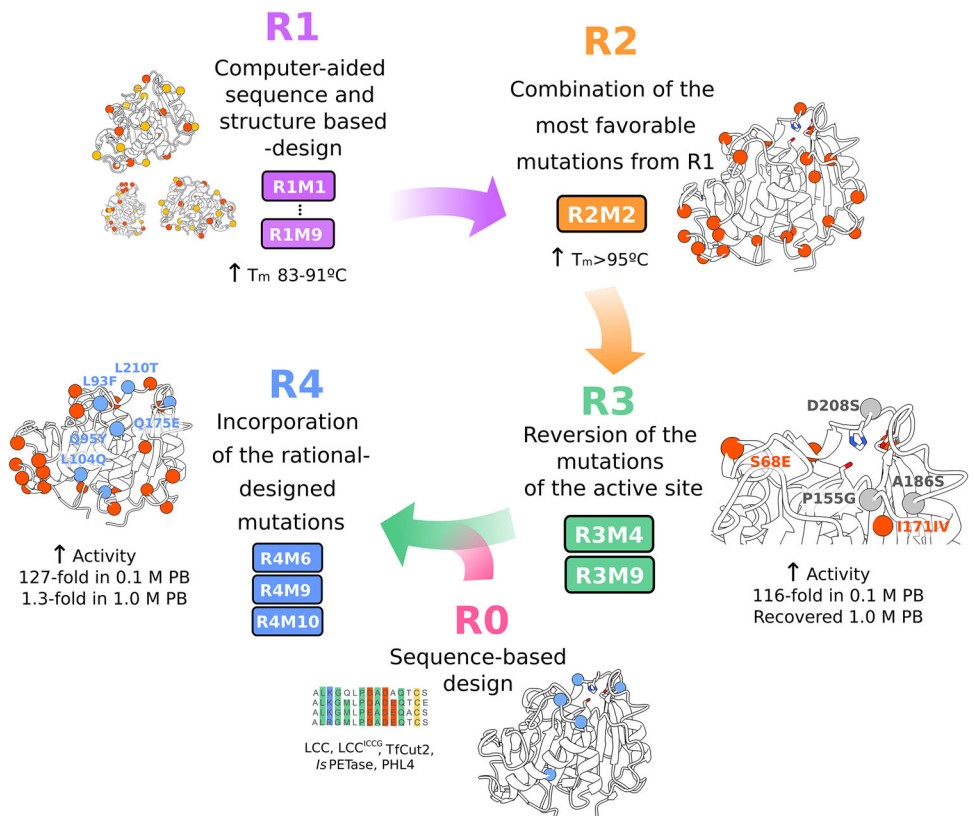

**Fig. 1 | Schema of PHL7 engineering workflow.** Round 1 (R1): Stability-enhancing mutations were engineered with the help of structure- and sequence-based design calculations using the Rosetta PROSS algorithm. Round 2 (R2): The most stability-enhancing mutations from R1 were combined into multi-point mutants, and their structure and function were characterized. Round 3 (R3): Mutations with activity-reducing effects near the active site were reverted to restore activity in 0.1 and 1.0 M phosphate buffers (PB). Round 4 (R4): Mutations identified in a preparatory round (round 0, R0) of sequence-based engineering were added to the best mutants from R3 to further enhance their activity.

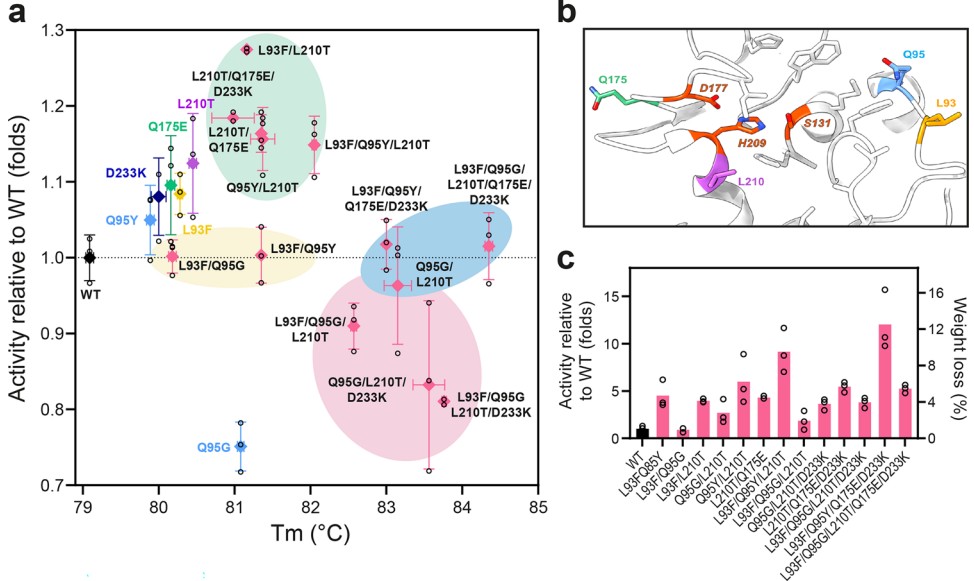

**Fig. 2 | Thermostability and PET-hydrolytic activity of PHL7 mutants carrying mutations near the active site. a** Activity measured at 70 °C and 1.0 M phosphate buffer versus their $T_m$ values. Reactions were carried out for 14 h using $1\,\mu g_{enzyme}\,mg_{PET}^{-1}$. The values from the variants L93F, Q95G, Q95Y, and D233K were taken from ref. 27. **b** Active site of PHL7 with the mutated residues indicated in colors, the catalytic triad depicted in red, and other binding pocket residues depicted in gray. The location of D233 on the opposite side of PHL7 is displayed in Supplementary Fig. 1c. **c** Activity at 70 °C and 0.1 M phosphate buffer of the PHL7 variants. Reactions were carried out for 14 h using $1\,\mu g_{enzyme}\,mg_{PET}^{-1}$. All measurements were conducted in triplicates ($n = 3$). Data are represented as black circles and mean ± standard deviation.

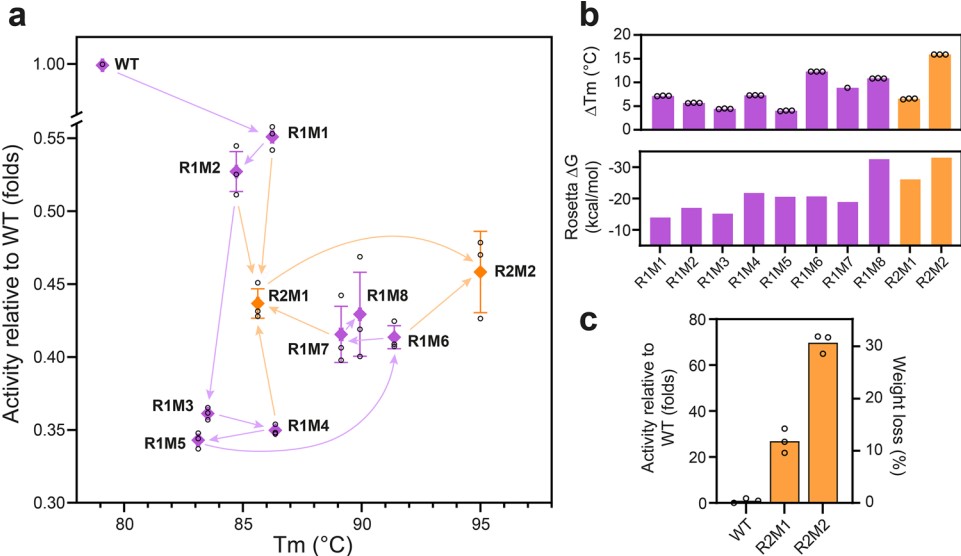

**Fig. 3 | Stability and PET-hydrolytic activity of PHL7 variants designed using the Rosetta PROSS algorithm. a** Fold-activity relative to the WT of R1 and R2 variants versus their $T_m$. Activity was measured at 70 °C and 1.0 M phosphate buffer. Note that the $T_m$ of R2M2 is >95 °C. The order in which mutations were introduced is indicated with arrows. **b** $T_m$ increase and Rosetta energy change of R1 and R2 mutants. **c** Fold activity of R2M1 and R2M2 in 0.1 M phosphate buffer at 70 °C relative to the WT (left Y-axis) and PET film weight loss under the same conditions (right Y-axis). All experiments were conducted in triplicates ($n = 3$). $T_m$ and activity data are represented as black circles and mean ± standard deviation.

L93F/L210T variant. Moreover, the presence of Q95Y decreases the activity of L93F/Q95Y/Q175E/D233K compared to L93F/Q175E/D233K. This suggests that while Q95Y alone enhances activity, its benefits may be diminished in specific mutational contexts.

Contrary to the findings of Pfaff et al.[28], where L93F/Q95Y (termed PES-H1 L92F/Q94Y in their study) was reported to have a 2.3-fold higher activity than the WT PHL7 at 72 °C, we observed no significant activity increase of this variant at 70 °C. This may be explained by the different reaction temperatures used and the higher stability of L93F/Q95Y over the WT. A steeper decline of the half-life and activity of the WT compared to L93F/Q95Y at 72 °C could have resulted in the observed activity differences.

At 65 °C and 1 M phosphate buffer, a similar trend was observed as at 70 °C (Supplementary Fig. 1a). L93F/L210T and L210T/Q175E/D233K displayed the highest PET degradation activities of 1.25-fold and 1.24-fold relative to the WT, corresponding to PET weight losses of 70.0% and 69.3%, respectively. Notably, L93F/Q95G/L210T/Q175E/D233K showed an increased activity of 1.16-fold (64.4% PET weight loss) compared to 70 °C, where its activity matches that of the WT PHL7 (Fig. 2a and Supplementary Fig. 1a).

With 0.1 M phosphate buffer and 70 °C, most mutants exhibit very low PET-hydrolytic activity, not exceeding 5% PET weight loss within 14 h of reaction. Notable exceptions are variants L93F/Q95Y/L210T and L93F/Q95Y/Q175E/D233K, which showed 8.2 and 12% PET weight loss, respectively (Fig. 2c). At 65 °C and in 0.1 M phosphate buffer, activities were notably higher than at 70 °C (Supplementary fig. 1a). Particularly noteworthy are variants L93F/Q95Y/L210T and Q95Y/L210T with 45.6 and 44.6% PET weight loss, respectively, which corresponds to a >15-fold higher activity than the WT (3.0% weight loss) (Supplementary Fig. 1a). Additionally, L210T/Q175E/D233K, L93F/Q95G/L210T/Q175E/D233K, and L93F/Q95G/L210T/D233K variants displayed increased activities, with respective fold changes of 12.1 (35.8% weight loss), 10.6 (31.3% weight loss), and 10.4 (30.9% weight loss) (Supplementary Fig. 1b). Interestingly, the variant L93F/Q95Y also exhibited higher activity in 0.1 M phosphate buffer at 60 °C (26.5% weight loss) compared to 70 °C (4.7% weight loss) (Fig. 2c and Supplementary Fig. 1b) confirming a previous report[7].

In summary, the initial round of PHL7 engineering identified activity-enhancing mutations, most notably L93F, Q175E, L210T, and D233K, which, upon combination, produced several multi-point variants with moderately improved activity and stability. However, the PET-hydrolytic activity in 0.1 M phosphate buffer remained at very low levels, requiring a further stabilization of PHL7.

## Computational design of PHL7 substantially increases its stability

To increase the stability of PHL7, we applied the Rosetta PROSS computational design algorithm[31], which integrates atomistic energy calculations with phylogenetic information to identify stabilizing mutations. We deliberately avoided the introduction of artificial disulfide bonds (e.g., between positions 205 and 251)[10], as these can reduce catalytic activity and negatively impact folding efficiency and recombinant expression[12]. Instead, we pursued a mutagenesis-based stabilization strategy that enhances thermostability while maintaining enzymatic performance and manufacturability.

In the first design round (R1), we generated nine different PHL7 variants, termed R1M1 to R1M9. The number of mutations increased stepwise from 8 mutations in R1M1 to 40 mutations in R1M9 (Supplementary table 1). In each step, the mutations in the respective previous variant were fixed, and between two and nine extra mutations were added. All PROSS-designed variants could be expressed as soluble proteins in *Escherichia coli*, and their stability and PET hydrolysis activity were determined. Only R1M9 exhibited an altered SEC profile and a significantly lower $T_m$ of 58.3 °C, indicating that its structure was significantly perturbed, which prohibited further characterization of this variant.

All R1 variants (except R1M9) showed an increased stability indicated by elevated apparent $T_m$ values compared to the WT PHL7 ($T_m$ = 79.1 °C) (Fig. 3a, b). The increase in $T_m$ was roughly proportional to the number of mutations. R1M6 displayed the highest $T_m$, exhibiting a 12.3 °C increase compared to the WT, followed by R1M8 with a 10.8 °C increase, and R1M7 with a 10.1 °C improvement. R1M4 and R1M1 displayed similar $T_m$ increases of 7.3 °C and 7.2 °C, respectively. The smallest enhancements were observed for R1M2, R1M3, and R1M5, with

increments of 5.6 °C, 4.4 °C, and 4.0 °C, respectively (Fig. 3a). The elevated $T_m$ for all R1 variants can be explained by their higher energetic stability as evaluated by the Rosetta energy function (Fig. 3b).

Subsequently, we determined the enzymatic activity of the R1 variants under the optimum conditions for PHL7 (1 M potassium phosphate buffer, pH 8.0 at 70 °C) and measured the extent of low-crystalline PET film degradation within 24 h of reaction. While all variants showed a loss of activity compared to the WT, R1M1 exhibited the least decline (0.55-folds, 70.5% weight loss), closely followed by R1M2 (0.53-folds, 67.4% weight loss). The variants R1M6, R1M7, and R1M8 exhibited lower values of 0.41-fold (52.8% weight loss), 0.42-fold (53.1% weight loss), and 0.40-fold (54.9% weight loss), respectively. The variants R1M3, R1M4, and R1M5 were the least active, with values of about 0.35-fold (44% weight loss) (Fig. 3a).

Since the PROSS algorithm selects mutations that should make additive contributions to stability, we reasoned that a combination of the energetically most favorable R1 mutations could lead to additional stability-enhanced versions of PHL7. Based on computational evaluation, we designed two second round (R2) variants, termed R2M1 and R2M2, by combining mutations from selected R1 variants that exhibited improved Rosetta energy scores relative to both the WT PHL7 and their respective predecessors in the design trajectory (Supplementary Fig. 3, Supplementary Methods). We note that R2M1 and R2M2 were designed purely based on computational inputs before testing the R1 variants. R2M1 contained all mutations from R1M1 (Q104L, R111T, T145P, E148K, G155P, T167V, L235T, F248P), two mutations from R1M2 (L176N, T219I), three from R1M4 (E68S, H109Y, S186A), and two from R1M7 (N113D, D196S). R2M2 contained all mutations from R2M1, except for L176N and T219I, and, in addition, five mutations from R1M6 (D36S, V115T, N118D, V171I, D216N), and six mutations from R1M9 (E6Q, Q34D, T91S, N161D, D198P, S208D).

When assessing the thermostability, R2M1 showed a $T_m$ increase of 6.5 °C relative to the WT, while R2M2 showed a striking increase of more than 16 °C (Fig. 3a, b). The exceptionally high $T_m$ of the R2M2 mutant of >95 °C was exceeding the detection limit of the nanoDSF device. In comparison, the polyester hydrolase BhrPETase from *Bacillus subtilis* exhibited the highest previously reported thermostability with a $T_m$ of 101 °C[36].

The activity of R2M1 and R2M2 on PET films under high-salt buffer conditions was comparable to that of the R1 mutants, showing a ca. 0.4-fold activity (51% weight loss) compared to the WT PHL7 (Fig. 3a). However, when tested at 0.1 M phosphate buffer concentrations, R2M2 displayed an activity increase compared to PHL7, degrading 30.7% of a low-crystalline PET film within 16 h, which corresponds to a 70-fold increase in activity relative to the WT (Fig. 3c). R2M1 also showed an 11.4-fold increase in PET-hydrolytic activity (Fig. 3c).

## Structural analysis of PHL7 R2M2 uncovers specific mechanisms for protein stabilization

With the aim of understanding the increased stability of R2M2 from a structural perspective, we determined its structure by X-ray crystallography with a resolution limit of 1.73–1.12 Å. The structures of R2M2 and PHL7 are highly similar, as indicated by a low Cα-atom RMSD of 0.60 Å over 258 residues (Fig. 4a), particularly considering that the two structures were determined in different crystal forms. The structural similarity is also very high for the substrate binding pocket. The residues in subsites I and II and of the catalytic triad (S131, D177, H209) align with an all-atom RMSD of 0.65 Å over 188 atoms (Fig. 4b). Thus, despite containing 24 mutations, R2M2 maintained sub-angstrom similarity to PHL7. This highlights the ability of the PROSS algorithm to predict mutations with a large net stabilizing effect, yet without modifying the active site structure or leading to large negative changes in the enzymatic activity.

The 24 mutations in R2M2 are distributed throughout the protein structure (Fig. 4c). The majority of them (21 mutations) are surface-exposed, and only three mutations occur at core positions. A closer inspection of the mutations reveals distinct stability-affecting mechanisms. Four mutations affect a cluster of Asp and Glu residues, changing them from acidic to neutral (E6Q, D196S, D198P) or acidic to basic residues (E148K). This acidic cluster is located on one side of PHL7, where it renders the electrostatic potential highly negative. Consistent with this, PHL7 has a low isoelectric point (pI) of 5.5, which increases to 6.2 in R2M2. In contrast, other PET hydrolases used for comparison in this study (e.g., ICCG) exhibit substantially higher pI values (8.5–9.5), reflecting their lower number of negatively charged residues (Supplementary Fig. 4). The four mutations in R2M2 significantly change the protein surface charge, rendering it more neutral (Fig. 4d), which could explain the stability gain of R2M2 under low salt concentrations. Studies on halophilic proteins, which carry a high number of acidic residues on their surface, like PHL7, show a protein-stabilizing effect of high salt concentrations due to surface charge compensation[37,38], which, however, may be less required for the stability of R2M2 due to its reduced negative charge density. To corroborate the role of these four mutations in PHL7, they were successively changed back to the original residues in PHL7, following the reverse order in which they were added in the R1 variants. The mutations Q6E, Q6E/D198P, Q6E/P198D/S196D, and Q6E/P198D/S196D/K148E in R2M2 led to a stepwise decrease of its PET-hydrolytic activity in 0.1 M phosphate buffer (Fig. 4e). However, in 1.0 M phosphate buffer, no significant activity change was found, indicating that the role of these mutations is linked to a charge-dependent effect. In addition, mutation E148K affects a sodium-binding site formed between E148 and D233 in PHL7. In contrast, in the R2M2 structure, no sodium ion is detected. Instead, K148 and D233 are positioned for an ionic interaction at a distance of 5.6 Å (Fig. 4f). Weak electron density for the terminal K148 CE and NZ atoms indicates alternate side-chain conformations. Consistent with this observation, MD simulations of R2M2 reveal multiple conformations, including a close salt-bridge interaction in ~45% of the frames (Supplementary Fig. 7), which likely contributes to the enhanced stability of R2M2.

Remarkably, one region of R2M2 contains four mutations in close proximity to each other–H109Y, R111T, N113D, and V115T. Each of these four mutations contributes to stability via a distinct mechanism (Fig. 4g). H109Y has a larger side-chain than its PHL7 counterpart, which optimizes the packing with neighboring hydrophobic residues (V30, F38, I43). R111T eliminates an unfavorable, electrostatic repulsion, which exists between R111 and R117 in PHL7, but is absent in R2M2. N113D undergoes a backbone displacement and side-chain reorientation, affecting the neighboring residues T112 and S114. As a consequence, S114 has flipped its position by 180° compared to the PHL7 structure and makes a helix-capping hydrogen bond interaction with L110 in R2M2, further enhancing stability. Furthermore, mutation V115T changes a surface-exposed, hydrophobic residue to a polar residue, enabling additional hydrogen bonds with water.

Three other mutations (G155P, S186A, V171I) are noteworthy because they are the only ones in the protein core and are next to the catalytic triad (Fig. 4h, Supplementary Fig. 5). G155P, V171I, and S186A improve core packing, and G155P may additionally increase backbone rigidity. Interestingly, G155P leads to the displacement of a water molecule and the positional shift of a second water molecule, which can be clearly defined in the X-ray structure of R2M2 due to its high resolution of 1.12 Å (Fig. 4h). Both waters are wired via a third water molecule to the catalytic D177. Interestingly, reversal of the G155P mutation, but not the S186A mutation, to the original amino acid in PHL7 leads to the reappearance of the described water molecule, which we characterized by determination of the respective high-resolution X-ray structures of R2M2-P155G, R2M2-A186S, and R2M2-P155G/A186S (Supplementary Figs. 5 and 6, Supplementary Table 4). This shows that the effect of G155P is linked to the displacement of a water molecule in addition to changes of the protein backbone

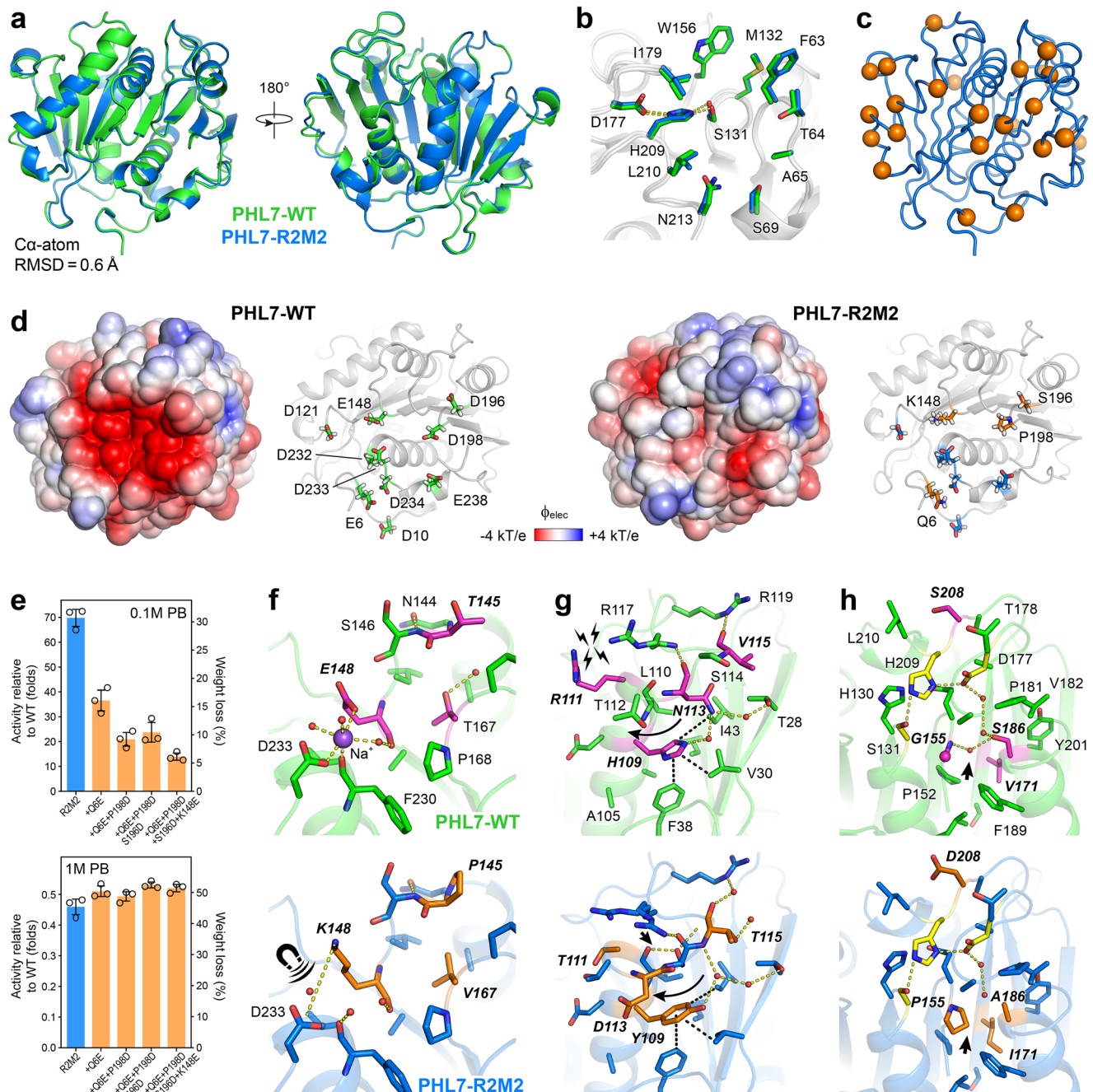

**Fig. 4 | Structural analysis of PHL7 R2M2. a** Crystal structure of PHL7 R2M2 (blue) superimposed on the structure of the WT PHL7 (green) (PDB 7NEI)[26]. The backbone trace is shown as cartoon diagram. **b** Superimposition of the PET binding pocket residues of the WT (green) and R2M2 (blue). The polar contacts connecting the catalytic triad residues (S131, D177, H209) are indicated by dashed lines. **c** Mutation sites distributed throughout R2M2 are shown as orange spheres. **d** Left: Electrostatic potentials of the WT and R2M2 mapped on their solvent-accessible surface. Right: A region of high negative electrostatic potential in the WT is formed by a cluster of ten acidic residues located on the same surface side. Mutations E148K, D196S, D198P, and E6Q neutralize the surface electrostatic potential of R2M2. **e** Fold-activity relative to the WT of R2M2 and R2M2 mutants at 70 °C and in 0.1 M phosphate buffer (upper) or 1.0 M phosphate buffer (lower) (left Y-axis) and PET film weight loss under the same conditions (right Y-axis). Experiments were

conducted in triplicates (*n* = 3). Data are represented as mean ± standard deviation. **f** Mutated residues T145P, E148K, and T167V close to the sodium binding site observed in the WT. The sodium ion-mediated contact between E148 and D233 in the WT is substituted by a salt bridge due to the E148K mutation. **g** Mutated residues H109Y, R111T, N113D, and V115T and their interactions in the WT (green) and R2M2 (blue). Polar and hydrophobic contacts are depicted as yellow and black dashed lines, respectively. An electrostatic repulsion between R111 and R117 in the WT, a helix capping hydrogen bond between L110 and S114 in R2M2, and a backbone shift for residues 112-115 in R2M2 are indicated by black arrows. **h** Mutated residues G155P, A186S, I171V, and D208S surrounding the catalytic triad in the WT (green) and R2M2 (blue), respectively. The absence of a water molecule next to G155P in the R2M2 structure is indicated by an arrow.

structure, which we have further investigated by MD simulations (see below). In summary, structural analysis of R2M2 revealed additive stabilizing effects of several designed mutations, establishing R2M2 as a promising template for the further engineering of PHL7.

## MD simulations uncover a critical role of mutations near the active site

Protein flexibility, in particular of the active site, has been recognized as an important factor for PET hydrolase activity and stability[39–41]. To

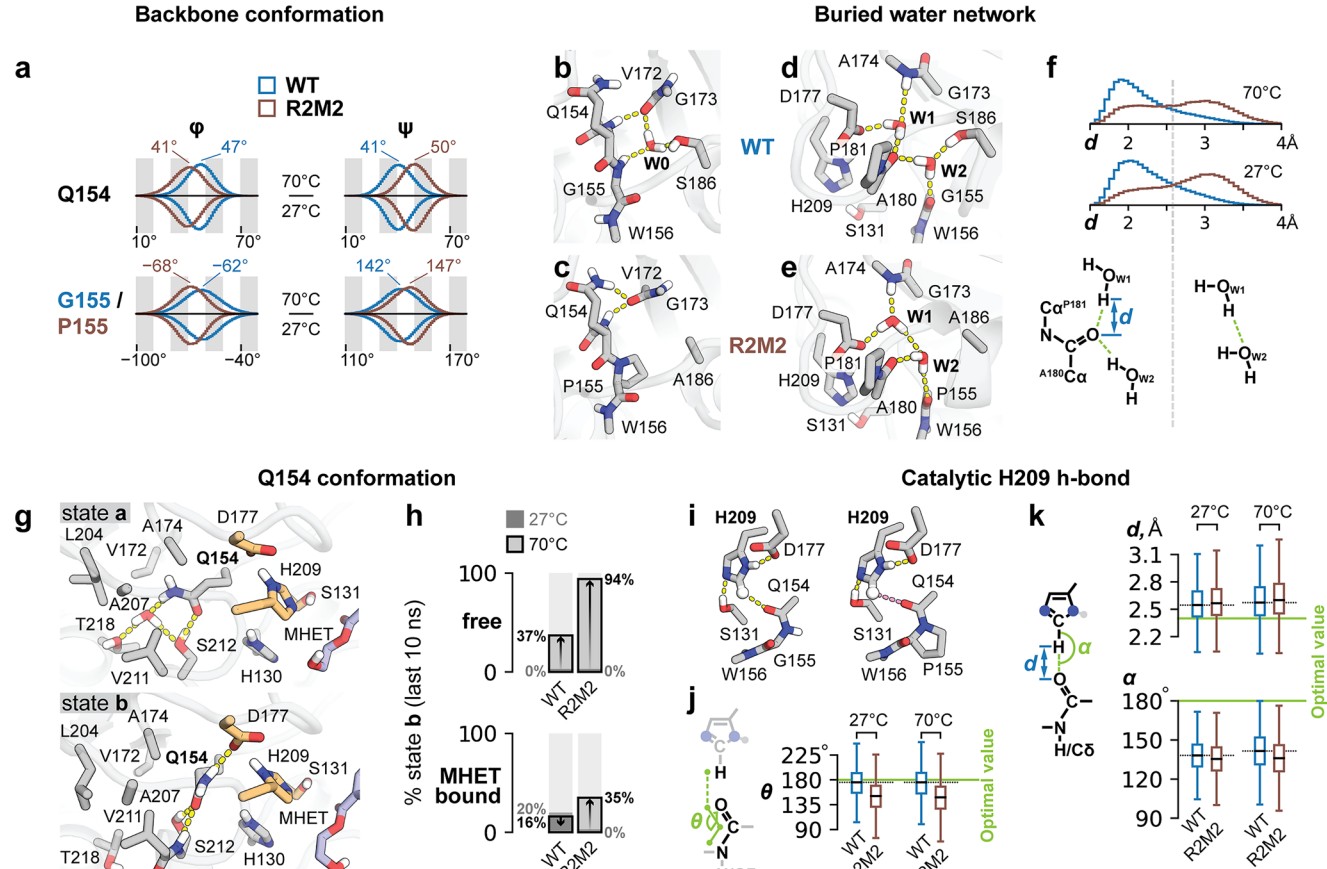

**Fig. 5 | MD simulation of R2M2. a** Different backbone conformation (Ramachandran angles – φ and ψ) of residues 154-155 in the WT PHL7 and R2M2. **b–e** Water network near the catalytic triad is restructured in R2M2. **f** Distance distribution between the W1 water molecule and the backbone carbonyl oxygen of A180 in the WT PHL7 and R2M2. **g** Side-chain conformation and hydrogen-bond interactions of Q154 in two major states identified in the MD simulations. **h** Proportions of alternative Q154 conformation in the WT and R2M2 in free and MHET-bound forms.

**i–k** Geometry of catalytic H209 Hε-OC H-bond. The distributions for the out-of-plane angle of the N-C = O•••H group, as well as the distance and angle of the C-H•••O hydrogen bond are shown. Box-and-whisker plots show the mean as the center, with the box representing the interquartile range and whiskers extending to the minimum and maximum values within 1.5 times the interquartile range. Shown is data over 5 replicates, 100 ns each.

investigate the dynamical features of R2M2 that might be tied to observable changes in the enzyme's activity and stability, we performed molecular dynamics (MD) simulations, modeling free and substrate-bound complexes of R2M2 and the WT PHL7 at 27 and 70 °C.

In the absence of the substrate molecule (MHET), the catalytic triad was observed to almost never adopt the catalytically competent arrangement at 27 °C, with the rise in the population of such arrangement occurring at an increased temperature of 70 °C (Supplementary fig. 8a). In non-reactive conformations, the catalytic S131 lost its hydrogen bond to the catalytic H209, and instead occupied the oxyanion hole or interacted with the non-catalytic H130. This observation suggests that the triad takes up the reactive arrangement mostly in the presence of the bound substrate, as found for other serine hydrolases[42]. Therefore, the structural features of the free enzyme might not reflect those of the substrate-bound active form, and we focused our study mostly on the substrate-bound complex.

The overall flexibility of R2M2 and the WT PHL7 was comparable (Supplementary Fig. 9a). However, considerable shifts in backbone conformations (φ, ψ) were found for particular residues (Fig. 5a and Supplementary Fig. 9b). The largest shifts were observed for residues 108−120, which were caused by the occurrence of 4 mutations in this region. As shown above (Fig. 4g), the mutations trigger a displacement of the cap of helix αB, leading to pronounced changes of the protein backbone. The backbone displacement associated with the N113D

mutation in the R2M2 structure and the resulting alterations in the interaction profile of residue S114 were maintained throughout the R2M2 MD simulation (Supplementary Fig. 10). In addition, radial distribution function analysis of the water shell surrounding residue 115 confirmed that the V115T substitution leads to an increase in local solvation of the R2M2 protein (Supplementary Fig. 11). Furthermore, Q154 and P155 (G155 in the WT), close to the active site, show distinct shifts of their φ/ψ distributions. These changes resulted in alterations in H-bonding networks, in which Q154 and P/G155 engage with buried water molecules and the catalytic D177 (Fig. 5b–d). The changes in R2M2 include a loss of a buried water molecule (W0 in Fig. 5b) and a change of the interaction pattern of a second buried water molecule (W1) (Fig. 5c). Whereas W1 bridges the catalytic D177 to the backbone of A180 in the WT, it is rewired in R2M2 to bridge D177 to another water molecule (W2) (Fig. 5c). In this way, the latter water molecule compensated for the loss of the hydrogen bond that occurred by the substitution of S186 to alanine. These features of R2M2 agree well with its observed lower activity than the WT.

The binding of the substrate was found to induce the formation of a minor occupied conformation of the Q154 side-chain at 27 °C in the WT PHL7 (Fig. 5g). This state was characterized by a switch of Q154 to form a hydrogen bond with the catalytic D177. This resulted in a slight shift in the conformation of the latter residue, which might have functional implications at the chemical stage (Supplementary Fig. 8b).

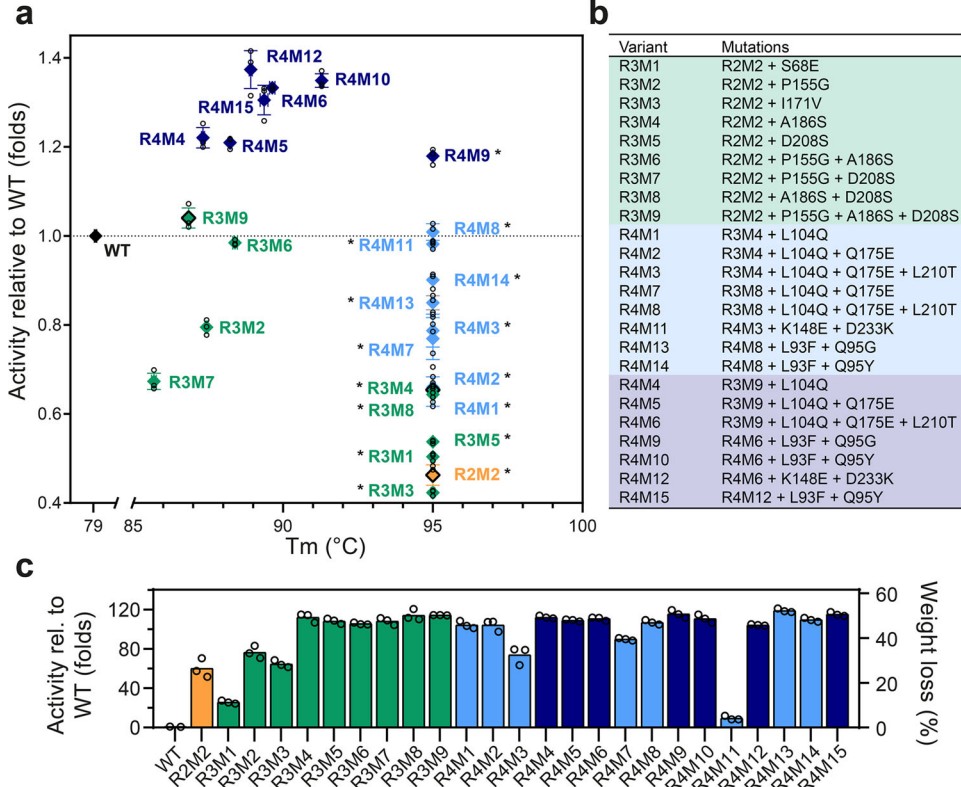

**Fig. 6 | Stability and PET-hydrolytic activity of R3 and R4 mutants derived from R2M2. a** Fold-activity relative to the WT PHL7 versus the $T_m$ values of R3 mutants (depicted in green) and R4 mutants (depicted in blue) at 1 M phosphate buffer concentration and 70 °C. R4 variants derived from R3M4 are depicted in light blue, and variants derived from R3M9 are depicted in dark blue. Note that the $T_m$ of variants marked with an asterisk is >95 °C. **b** List of variants with their

mutations and their names. **c** PET-hydrolytic activity measured in 0.1 M phosphate buffer at 70 °C for R3 and R4 mutants. The activities are indicated as relative to the WT (left Y-axis) and percentage of weight loss (right Y-axis). Experiments were performed in triplicates ($n = 3$). Data are shown as black circles and mean ± standard deviation.

While in the absence of the substrate at 27 °C, the alternative conformation of Q154 was absent, it became more populated at 70 °C and was even the major one for R2M2. We observed the same effects of the raised temperature also for the substrate-bound form of R2M2, but not for the WT. The temperature dependence could be linked to the hydrophobic environment in which Q154 is located, since it is known that hydrophobic interactions grow stronger with increasing temperature[43].

Notable changes were also observed for the conformation and interactions of the catalytic H209 (Fig. 5i–k). The CH-O hydrogen bond of H209 to the backbone of residue Q154 was shorter and closer to the optimal geometry (angle of 180°) in case of the WT enzyme, whereas it deviated more from the optimal geometry in case of R2M2. The G155P mutation in R2M2 has a direct influence on this hydrogen bond geometry. This CH-O hydrogen bond was previously identified to be important for all catalytic triads containing histidine, owing to the pronounced contribution it plays in achieving a functional catalytic triad geometry when the histidine becomes cationic in the course of the reaction[44]. Thus, our results highlight the crucial functional implications of the G155P substitution. Moreover, the interaction geometry between H209 and the catalytic S131 in R2M2 exhibited a greater deviation from the optimal geometry (Supplementary Figs. 12, 13), whereas the configurational parameters of the D177-H209 interaction remained essentially unchanged (Supplementary Figs. 14, 15). Altogether, our MD simulations allowed us to identify structural features linking sequence variations in R2M2 to notable shifts in conformations of the catalytic triad. These correlations might explain the observed lower catalytic activity of R2M2.

## Iterative engineering around the active site of R2M2 restores high PET-hydrolytic activity

In an effort to restore the high PET-hydrolytic activity of PHL7, which was diminished in the R2M2 variant, while maintaining the high stability of R2M2, we systematically reversed the mutations close to the active site of R2M2 to the original amino acids in PHL7. Mutations E68S, G155P, V171I, S186A, and S208D in R2M2 were reverted, first by one mutation at a time and afterwards in combinations, leading to the third round of variants (R3).

The R2M2 single mutants S68E (R3M1), I171V (R3M3), A186S (R3M4), and D208S (R3M5) exhibited all a $T_m$ value above 95 °C (Fig. 6a). In contrast, the P155G variant (R3M2) showed a reduced $T_m$ of 87.5 °C, but a noticeable increase in activity to 0.8-fold relative to PHL7 (78.6% weight loss) in 1.0 M phosphate buffer, almost doubling the activity of R2M2 under these conditions (0.46-fold, 45.7% weight loss) (Fig. 6a). The introduction of A186S and D208S also led to increased activities of 0.65-fold (64.3% weight loss) and 0.54-fold (53.2% weight loss), respectively, relative to PHL7 (Fig. 6a). Conversely, mutations S68E and I171V showed no notable improvement, exhibiting activities of 0.50-fold (49.8% weight loss) and 0.42-fold (41.9% weight loss), respectively.

At a 0.1 M phosphate buffer concentration and 70 °C, the enzymatic activity enhancements compared to the WT PHL7 were substantial. The increases were 25.7-fold (11.3% weight loss) for S68E, 76.3-fold (33.6% weight loss) for P155G, 64.6-fold (28.4% weight loss) for I171V, with the largest improvements being 111.9-fold (49.3% weight loss) for A186S and 108.2-fold (47.6% weight loss) for D208S (Fig. 6c).

Given the importance of residue 155 in R2M2, we further investigated this position in PHL7, replacing it with small amino acids (Pro,

Ala, Ser, Thr) present in homologous enzyme sequences. Substitution of G155 to Pro and Ala led to small $T_m$ increases of 2.5 °C and 1.0 °C, respectively, whereas mutations to Ser and Thr reduced the $T_m$ by 3.7 °C and 9.9 °C, respectively (Supplementary Fig. 16c). However, all mutations reduced the activity at 70 °C (Supplementary Fig. 16a, b), with the smallest decreases observed for PHL7-G155A (0.92-fold) followed by G155P (0.69-fold) and G155S (0.46-fold).

To evaluate potential synergistic effects, the most impactful single mutations—P155G, A186S, and D208S—were combined into double and triple variants. As shown in Fig. 6a, variants containing P155G exhibited reduced thermostability relative to R2M2, whereas the A186S/D208S combination (R3M8) retained high thermal stability. Notably, the triple variant R2M2-P155G/A186S/D208S (R3M9) demonstrated the highest catalytic activity (1.04-fold), comparable to that of PHL7 in 1.0 M phosphate buffer and at 70 °C.

The improvements in PET-hydrolytic activity were even more striking in 0.1 M phosphate buffer as shown in Fig. 6c. Regarding the single point mutants, R2M2-A186 (R3M4) was found to be the most active variant under these conditions, showing a 111.9-fold higher activity relative to the WT PHL7 (Fig. 6c). This was followed by R2M2-D208S (R3M5), which showed a 108.1-fold activity improvement. However, the mutations R2M2-S68E (R3M1) (25.7-fold), R2M2-P155G (R3M2) (76.3-fold) and R2M2-I171V (R3M3) (64.6-fold) did not further improve activity at low buffer concentrations (Fig. 6c). The combination of single mutations in the variants R2M2-P155G/A186S/D208S (R3M9), R2M2-A186S/D208S (R3M8), R2M2-P155G/D208S (R3M7), and R2M2-P155G/A186S (R3M6), led to activity enhancements of 114.2-fold, 113.9-fold, 108.1-fold, and 105.5-fold, respectively, relative to the WT (Fig. 6c).

In summary, the most promising variants after this iterative engineering phase were the R2M2 triple mutant P155G/A186S/D208S (R3M9), having high activity at both buffer conditions as well as high thermostability (≥7.8 °C relative to the WT), and the R2M2 single mutant A186S (R3M4), exhibiting an even higher thermostability (≥16 °C relative to the WT) and the highest activity determined in 0.1 M buffer. Thus, we subjected these two variants to a final engineering round aimed at further boosting their PET-hydrolytic activity.

### Introduction of rationally designed mutations into R3M4 and R3M9 generates variants with further enhanced activity

The mutations L93F, Q95Y, Q95G, Q175E, L210T, and D233K, characterized in the first design step, were introduced in the best R3 mutants. R3M4 was chosen as a template because it showed the best tradeoff between activity and thermostability. R3M9 was selected because of its high activity that surpassed the activity of the WT enzyme under high-salt buffer conditions. In addition, mutation L104Q was added to change one of the PROSS-designed mutations (Q104L) back to the original Gln residue from PHL7, because analysis of the R2M2 X-ray structure indicated a loss of hydrogen bonds due to Q104L. The variants generated in the final engineering round were designated round 4 (R4) variants, and their codes and individual mutations are summarized in Fig. 6b.

All variants derived from R3M4 retained high thermostability (>95 °C; Fig. 6a, Supplementary Fig. 2). Among these, progressive combinations of L104Q, Q175E, D208S, and L210T led to the most pronounced enhancement in activity, with variant R4M8 reaching WT levels (1.01-fold). Additional modifications of R4M8, including variant R4M11, which carried a salt bridge-swapping double mutation D233K/K148E, and variants R4M13 (L93F/Q95G) and R4M14 (L93F/Q95Y), failed to yield further improvements.

In the R3M9 background, stepwise introduction of L104Q, Q175E, and L210T produced variants with progressively higher stability and activity, culminating in R4M6, which exhibited the strongest enhancement relative to the WT (1.33-fold activity). Subsequent introduction of the salt bridge-swapping mutation D233K/K148E

produced variant R4M12, which retained a similar $T_m$ and activity (1.37-fold). Positions 93 and 95 were also investigated in the R4M6 template by introducing the mutations L93F/Q95G and L93F/Q95Y, resulting in variants R4M9 and R4M10, respectively. R4M9 showed a substantially increased $T_m$ (>95 °C), whereas R4M10 exhibited modest $T_m$ and activity increases compared to R4M6. Overall, R4M6, R4M10, and R4M12 emerged as the top-performing variants from this round, providing the most favorable trade-off between catalytic efficiency and thermostability (Fig. 6a).

Under low-salt buffer conditions, the R4 variants were considerably more active than the WT (Fig. 6c). Among the variants derived from R3M9, R4M9 and R4M15 exhibited the highest activity, with 115.3- and 115.1-fold increases relative to the WT, respectively. R4M4 and R4M6 also displayed notable improvements, achieving 112.1- and 110.6-fold increases, respectively, equivalent to R4M10, which also showed a 110.6-fold increase relative to the WT. R4M5 and R4M12 demonstrated slightly lower, yet substantial, enhancements in activity, with 108.9- and 104.2-fold increases, respectively. The variants derived from R3M4 exhibited a lower activity compared to the variants from the previous round, with the exception of R4M13, which achieved a 118.8-fold increase in activity compared to the 111.9-fold increase observed with R3M4. Interestingly, R4M11, in which the salt-bridge residue order is reversed relative to R4M8 (E148–K233 instead of K148–D233), displayed activity comparable to R4M8 in high-salt buffer but showed a drastic loss of activity in 0.1 M phosphate buffer. The salt bridge E148-K233 contained in this variant may not have formed effectively in the low-salt buffer, indicating that the activity of the enzyme is critically dependent on the ionic strength of the medium.

To directly assess the stabilizing effect of phosphate ions, apparent $T_m$ values of PHL7 and selected R4 variants were determined across a range of phosphate buffer concentrations (Supplementary Fig. 17). The results revealed a progressive increase in thermal stability, reaching $T_m$ values above 95 °C for all variants at the highest concentration of 1.0 M.

### Comparison of PHL7 R4 variants with top-tier reference PET hydrolases

The PET-degrading performance of the best R4 variants was subsequently evaluated across different temperatures (65, 70 °C) and phosphate buffer concentrations (0.1, 1.0 M) in comparison with the WT PHL7, HotPETase, ICCG, LCC-A2, and TurboPETase. Electrochemical Impedance Spectroscopy (EIS) was employed to measure the degradation rate of semi-crystalline PET films in μm/h as described previously[45]. At 70 °C in 1 M buffer, all R4 variants exceeded the performance of the reference enzymes WT PHL7, HotPETase, ICCG, and LCC-A2, which exhibited mean degradation rates of 15.7 μm/h, 3.0 μm/h, 12.3 μm/h, and 16.0 μm/h respectively, but remained slightly below that of TurboPETase (23.1 μm/h) (Fig. 7a). R4M6 exhibited the highest degradation rate of 21.0 μm/h, corresponding to a 1.3-fold increase over the WT PHL7. R4M12 showed a similar performance with a mean degradation rate of 20.4 μm/h. R4M9 displayed a degradation rate of 19.3 μm/h (1.2-fold increase), while R4M15 surpassed the WT PHL7 with a rate of 18.0 μm/h (1.1-fold increase). In contrast, R4M10 showed only a modest increase with a mean activity of 16.6 μm/h, representing a 6% improvement compared to the WT (Fig. 7a). At 65 °C in 1 M buffer, a similar trend with overall lower degradation rates was observed. All R4 variants outperformed the reference enzymes HotPETase, ICCG, and LCC-A2, which showed degradation rates of 2.7 μm/h, 6.3 μm/h, and 10.4 μm/h, respectively, while TurboPETase reached 14.4 μm/h. The mean activity of the WT PHL7 of 10.5 μm/h was surpassed by R4M6, R4M10, R4M12, and R4M15, which displayed degradation rates of 13.0 μm/h, 12.1 μm/h, 12.0 μm/h, and 13.0 μm/h, respectively. R4M9 showed a slightly lower activity of 9.5 μm/h compared to the WT (Fig. 7a).

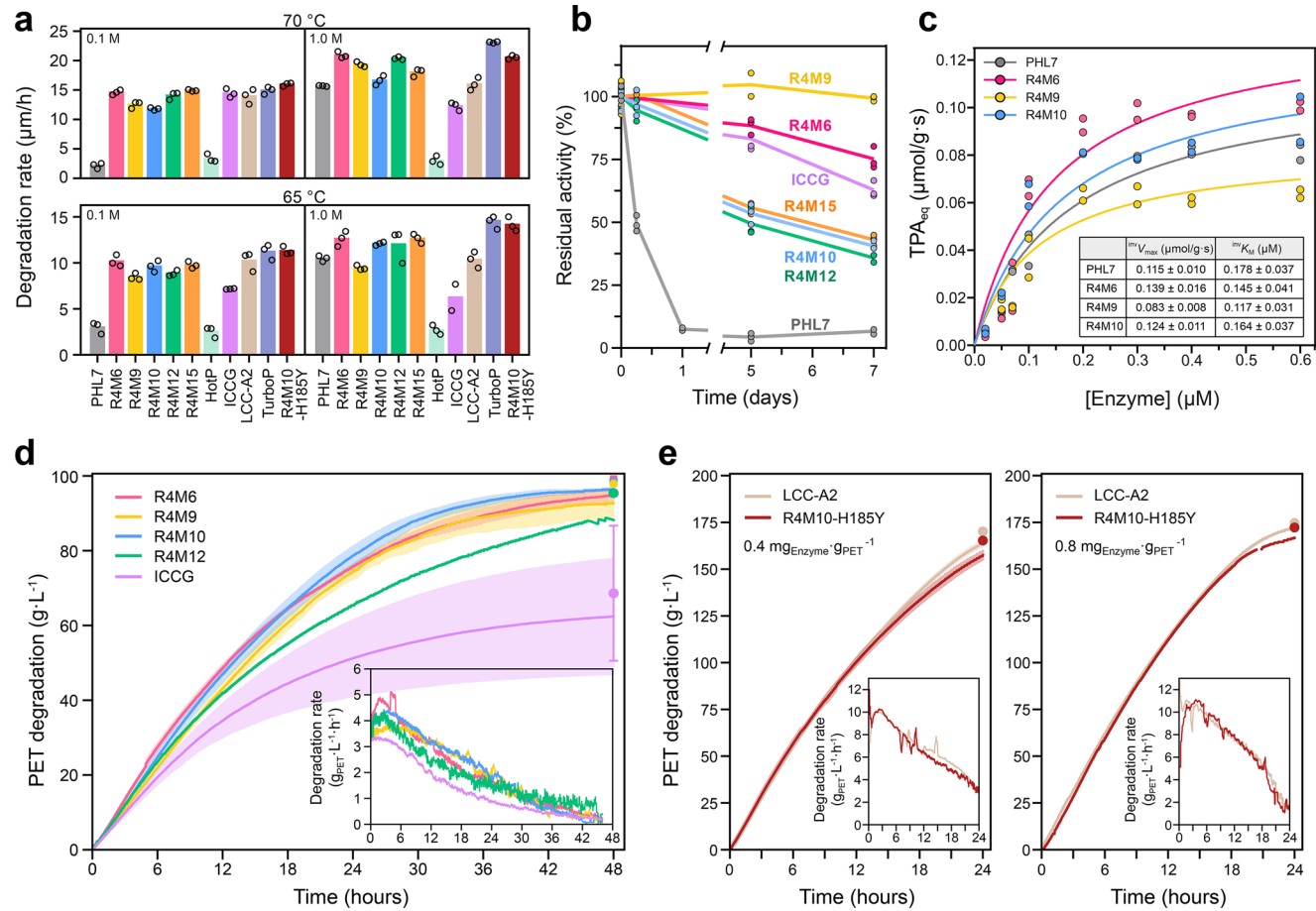

**Fig. 7 | PET film degradation by PHL7 R4 mutants and reference enzymes. a** PET film degradation rates measured by EIS in μm per h at 70 and 65 °C in 0.1 M or 1 M phosphate buffer. PHL7, its R4 variants, and R4M10-H185Y are compared to ICCG, HotPETase (HotP), LCC-A2, and TurboPETase (TurboP). Data are represented as mean ± SD ($n = 3$ independent experiments). **b** Residual activity of the R4 variants, PHL7, and ICCG after incubation at 65 °C for 0.5, 1, 5, and 7 days. The activity was determined using PCL as substrate ($n = 3$ independent experiments). **c** Inverse Michaelis-Menten plots showing the rate of TPA release from PET determined after 4 h of reaction as a function of enzyme concentration for PHL7, R4M6, R4M9, and R4M10, respectively. Experiments were performed in independent duplicates ($n = 2$). Lines represent the best fit to the equation $v = {}^{inv}V_{max} \cdot [E] / {}^{inv}K_M + [E]$. The derived kinetic parameters are displayed in the inset table. **d** Low-crystalline PET flake degradation (10% w/w) by R4 variants and ICCG at a concentration of

0.8 mg$_{enzyme}$ g$_{PET}^{-1}$ in a pH-controlled bioreactor. The activity was calculated by determining the consumption of NaOH during the reaction (solid line) and by measuring the residual PET weight after stopping the reaction (point marker). Experiments were performed in independent duplicates ($n = 2$) for the R4 variants and in quintuplets ($n = 5$) for ICCG. The solid line represents the mean, and the shaded area (or error bar) represents ±1 standard deviation. The inset graph shows the PET film degradation rate (in g$_{PET}$ L$^{-1}$ h$^{-1}$) of the R4 variants and ICCG. **e** Degradation of 20% (w/w) PET by LCC-A2 and R4M10-H185Y in a pH-controlled bioreactor measured using NaOH consumption (solid line, mean; shaded area, ±1 SD) or residual PET weight (point marker). Experiments were performed using an enzyme concentration of 0.4 mg$_{enzyme}$ g$_{PET}^{-1}$ (left, $n = 2$ independent experiments) or 0.8 mg$_{enzyme}$ g$_{PET}$-1 (right, $n = 1$). The inset graphs show again the PET film degradation rate.

At 70 °C in 0.1 M buffer, WT PHL7 exhibited a mean degradation rate of 2.3 μm/h, notably less compared to ICCG, LCC-A2, and Turbo-PETase, which showed rates of 14.4 μm/h, 13.9 μm/h, and 14.9 μm/h, respectively. The R4 variants showed activities comparable to those of the best benchmark enzymes under these conditions (Fig. 7a). At 65 °C in 0.1 M buffer, the R4 variants exceeded the performance of WT PHL7 (3.0 μm/h), HotPETase (2.5 μm/h), and ICCG (7.2 μm/h), and were only marginally less active than LCC-A2 (10.3 μm/h) and TurboPETase (11.2 μm/h). Among the R4 designs, R4M6 emerged as the most active variant (10.2 μm/h), followed by R4M15 (9.8 μm/h), R4M10 (9.6 μm/h), R4M12 (8.6 μm/h), and R4M9 (8.4 μm/h) (Fig. 7a).

To evaluate long-term stability, a critical factor for efficient PET degradation, we incubated the enzymes over 1 week at 65 °C in 0.1 M buffer and measured their residual activity. PHL7 lost 50% of its activity within the first 6 h, and its activity was completely depleted after 1 day, underscoring its poor thermal stability and critical buffer dependence under elevated temperatures. In contrast, R4M10, R4M12, and R4M15

retained 50% of their activity for 5 days, highlighting their enhanced stability compared to the WT PHL7. R4M9 stood out as the most stable variant, maintaining 99% of its activity within 1 week of incubation. This exceptional performance was followed by R4M6, which retained 75%, and ICCG, which retained 63% of its activity. R4M15, R4M10, and R4M12 retained 43, 40, and 36% of their initial activity, respectively (Fig. 7b). These findings highlight the superior stability of R4M9 and R4M6, making them promising candidates for applications in bioreactor processes.

We further analyzed the kinetic differences of the R4 variants by the inverse Michaelis-Menten approach[46] (Fig. 7c). The $^{inv}K_M$ values of R4M6 and R4M10 were 0.145 ± 0.041 μM and 0.164 ± 0.037 μM, respectively, whereas those of R4M9 and the WT PHL7 were 0.117 ± 0.031 μM and 0.178 ± 0.037 μM, respectively (Fig. 7c inset), indicating that R4M9 displayed the highest affinity to the PET substrate. Overall, the $^{inv}K_M$ values of the three R4 variants were very similar and smaller than that of PHL7. Furthermore, the $^{inv}V_{max}$ values

of R4M6, R4M10, and R4M9 were $139 \pm 16 \, \text{nmol g}^{-1} \text{s}^{-1}$, $124 \pm 11 \, \text{nmol g}^{-1} \text{s}^{-1}$, and $83 \pm 8 \, \text{nmol g}^{-1} \text{s}^{-1}$, respectively (Fig. 7c inset).

## The R4 variants of PHL7 outperform ICCG in PET degradation under industrial process conditions

To evaluate the performance of the R4 variants at industrially relevant conditions using high substrate concentrations, we investigated the degradation of 10% (w/w) low-crystalline PET flakes in a pH-controlled bioreactor at 65 °C in 0.1 M potassium phosphate buffer, pH 8.0, and an enzyme concentration of $0.8 \, \text{mg}_{enzyme} \, \text{g}_{PET}^{-1}$. R4M6 and R4M10 performed particularly well, degrading $75.7 \pm 0.7\%$ and $78.7 \pm 2.5\%$ of PET, respectively, within 24 h, compared to $51.2 \pm 17.0\%$ achieved by ICCG (Fig. 7d). R4M10 surpassed all other variants and degraded 90% of the PET in 31.8 h. The variant R4M6 achieved 90% degradation in 37.1 h, which was comparable to the time needed by R4M9 (38.0 h) to reach the same extent of PET degradation. In contrast, R4M12 showed a lower performance, achieving 65.3% PET degradation in 24 h and 88.7% within 48 h of reaction. However, all R4 variants exceeded the performance of ICCG, which achieved 62.4% degradation within 48 h. These results indicate that R4M6 and R4M10 performed excellent under high substrate concentration conditions, exhibiting the highest PET-hydrolytic activity of the engineered PHL7 variants. Both variants showed a maximum production rate of about $4.5–5.0 \, \text{g}_{hydrolyzed \, PET} \, \text{L}^{-1} \text{h}^{-1}$ compared to about $3.2 \, \text{g}_{hydrolyzed \, PET} \, \text{L}^{-1} \text{h}^{-1}$ achieved by ICCG (Fig. 7d inset).

The strong performance of R4M10 in the PET degradation bioreactor experiment indicated its suitability as a promising candidate for enzymatic PET recycling processes. To assess its capacity for further engineering and its robustness at elevated substrate loadings, we introduced one additional mutation, H185Y—recently reported to enhance the activity of WT PHL7 in FlashPETase[30]—and evaluated the resulting R4M10-H185Y variant using EIS. The H185Y substitution increased the activity of R4M10 by 17−35%, demonstrating that the R4M10 scaffold can accommodate additional beneficial mutations. In 0.1 M buffer at 65 and 70 °C, R4M10-H185Y performed on par with the top-tier benchmark enzymes LCC-A2 and TurboPETase, and in 1.0 M buffer, its activity was only marginally lower than that of TurboPETase (Fig. 7a).

In a pH-controlled bioreactor containing 20% (w/w) low-crystallinity PET at 68 °C in 0.1 M potassium phosphate buffer, R4M10-H185Y exhibited comparable performance to LCC-A2, which was selected as the benchmark due to its superior expression in *E. coli*. At an enzyme loading of $0.4 \, \text{mg}_{enzyme} \, \text{g}_{PET}^{-1}$, R4M10-H185Y and LCC-A2 achieved 79% and 82% PET degradation, respectively, within 24 h (Fig. 7e). Doubling the enzyme concentration to $0.8 \, \text{mg}_{enzyme} \, \text{g}_{PET}^{-1}$ resulted in only modest increases (84% and 86% for R4M10-H185Y and LCC-A2, respectively), indicating that enzyme saturation had been reached. Taken together, these results demonstrate that our best R4 variants can sustain high PET degradation rates under industrially relevant, high-substrate-loading conditions while requiring comparatively low enzyme quantities, underscoring their potential for scalable enzymatic PET recycling.

## X-ray crystallography and MD analyses of R4 variants reveal molecular determinants for improved PET binding

Considering the increased activity of the R4 variants, we aimed to analyze the structural and dynamic features that underlie their functional characteristics. We determined the X-ray structures of R4M6 and R4M10 to resolutions of 1.53 Å and 1.10 Å, respectively. The two structures are almost indistinguishable to each other (Cα-atom RMSD = 0.08 Å) and highly similar to the structure of R2M2 (Cα-atom RMSD = 0.33 Å) (Fig. 8a). The slightly larger RMSD value for R2M2 is likely due to the different crystal form of R2M2 (P2₁) compared to R4M6 and R4M10 (P6₁22). The aforementioned K148−D233 salt bridge is observed in the R4M6 and R4M10 structures with well-defined

electron density and persisted in MD simulations (Supplementary Fig. 7), whereas it is less clearly resolved in the R2M2 structure. This observation may reflect different crystallization conditions, as R2M2 was crystallized at pH 4.0, where partial protonation of D233 could weaken the ionic interaction.

The mutations that distinguish R4M6 and R4M10 from R2M2 are all located in the vicinity of the substrate binding pocket (Fig. 8b). To analyze their effect on PET binding, we performed extensive Hamiltonian replica exchange (HREX) simulations of PET substrate (4× MHET units) binding, using the structures of PHL7, R4M6, R4M10, and a model of R4M9 as inputs. For each enzyme, we observed an extremely diverse ensemble of bound poses (Fig. 8c, Supplementary Fig. 19a–c). In R4M10, Q95Y substitution induced unique substrate conformations due to steric clashes with PET rings. L210T was found to have the most dramatic effect in shaping the distribution of bound substrate conformations (Supplementary Fig. 19d), increasing the diversity in engineered variants. Our results show that PET hydrolases assume not only one or a few dominant enzyme-substrate complex conformations, but rather a wide ensemble. However, it remains to be elucidated how individual substrate conformations influence the efficiency at the chemical stage of the reaction.

The R4 variants were found to show a higher propensity for substrate binding near the active site than the WT PHL7 (Fig. 8d), consistent with the $^{inv}K_M$ data. R4M6 has a dramatically increased proportion of these states. However, most of them remained non-reactive due to difficulties in entering the oxyanion hole (Fig. 8e). R4M9 and R4M10 alleviated this hindrance, which resulted in a much higher proportion of pre-reactive states overall, in which the carbonyl oxygen of the ester bond is hydrogen-bonded to the NH donors of the oxyanion hole. Interestingly, R4M9 exhibited a markedly different side-chain conformational distribution of M132 (Supplementary fig. 19e–h). This residue forms a binding site for the attacked PET unit, which might present a connection to the lower R4M9 activity in the experiment.

Molecular Mechanics Poisson-Boltzmann Surface Area (MMPBSA) analysis showed that the R4 variants displayed higher interaction energies than the WT (more positive values indicate lower binding strength; Supplementary Table 5), suggesting that their enhanced catalytic activity is unlikely to result from increased substrate binding affinity. Rather, it may arise from a greater population of pre-reactive states, improved active-site organization, as discussed above, or other factors such as a reduced activation energy barrier. We further note that the entropic contribution to the binding free energy, associated with reorganization of the enzyme's binding pocket and the substrate, cannot be reliably quantified using the MMPBSA approach and may differ substantially among variants.

Closer examination of simulation results revealed a role of F63 conformation in the observed binding differences. We found that the F63 side-chain can occlude the catalytic triad (Fig. 8f, g), preventing the correct positioning of the substrate, and that this behavior depends on the substitutions in engineered variants. R4M10 showed a significantly diminished proportion of such self-inhibited states, which aligned well with its better activity (Fig. 8h). In R4M10, the loop 93-95 was more flexible while the catalytic site remained reaction-ready, which might facilitate easier early binding events for longer PET chains (Fig. 8g).

Overall, the simulation results revealed unique features that might allow R4M10 to achieve its high level of activity. They also suggest that a closer attention should be paid to residues 93−95 in further engineering campaigns due to their intricate and pronounced effects on the structure and dynamics of PET hydrolases.

## Discussion

In the past few years, PET hydrolases with improved activity and thermostability were developed using different protein engineering

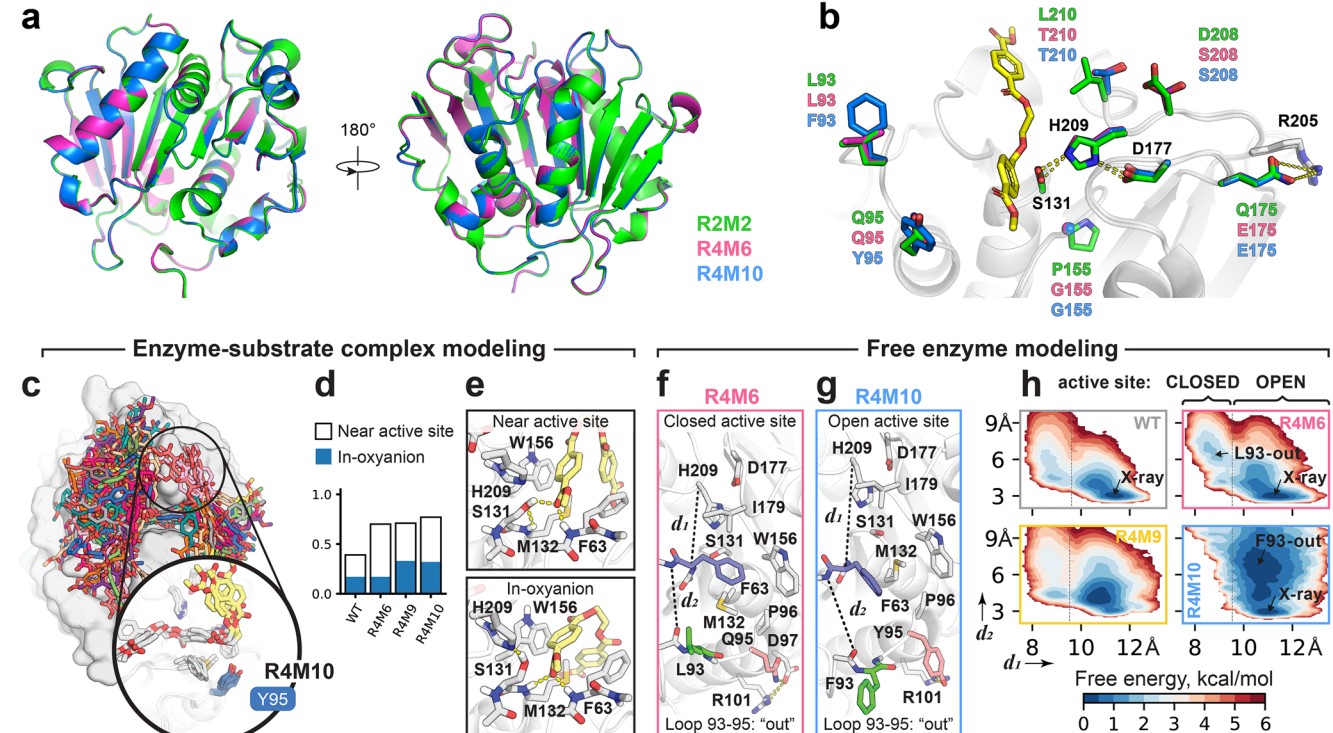

**Fig. 8 | Structure and MD simulation analysis of R4M6, R4M9, and R4M10.**
**a** Crystal structures of R4M6 (magenta) and R4M10 (blue) superimposed onto the structure of R2M2 (green). **b** Superimposition of the PET binding pockets of R2M2, R4M6, and R4M10. The amino acid residues which are different in R4M6 and R4M10 compared to R2M2 are shown as sticks and are labeled. Polar contacts between the catalytic triad residues (S131, D177, H209) and between E175 and R205 are indicated by dashed lines. A PET dimer model is depicted in yellow. **c** Overview of enzyme-substrate complex conformations sampled with HREX. R4M10 displays a unique direction of the PET chain due to the Q95Y substitution. **d** Efficiency of variants at binding the substrate in a catalytically-competent conformation. **e.**

Distinction between a bound, but not pre-reactive ("Near active site"), and a pre-reactive ("In-oxyanion") enzyme-substrate complex state. The "near active site" state was defined as having a OG(S131)-CO(MHET) distance of <5.5 Å and a N(M132)-O(MHET) distance between 5 and 7.5 Å. The "in-oxyanion" state was defined by a N(Met132)-O(MHET) distance of <5 Å. **f–h** Focused MD investigation of binding site conformations. **f** Inactive conformation with F63 occluding the catalytic site ("closed"). **g** Active conformation with an arrangement of residues similar to the crystal structure ("open"). **h** Free energy profiles show that R4M10 has a lowered propensity to assume an inactive closed form, while at the same time having an increased flexibility of a loop harboring L93F substitution.

strategies such as rational design[10–12,20,22], directed evolution[15,17,18], or machine learning-assisted protein design[16,19,24]. Here, we used a computer-assisted, semi-rational approach to engineer the PET hydrolase PHL7 for more efficient PET degradation under low-salt buffer and high-temperature conditions. Guided by calculations with the Rosetta PROSS algorithm[31], up to 24 mutations were installed into PHL7, which improved its thermostability and diminished the requirement for high-salt buffers at elevated temperatures, yielding a very stable variant (R2M2) as template for further engineering rounds. Subsequent introduction of several rationally designed mutations near the active site further enhanced enzyme activity, resulting in variants with a >110-fold higher activity than the original PHL7 in 0.1 M buffer at 70 °C. The best variants outperform other engineered hydrolases, such as HotPETase[15] and ICCG[12] in terms of their long-term stability and extent of PET degradation in reactor systems using industrially relevant high substrate concentrations. The engineered PHL7 variants include many novel mutations that, to the best of our knowledge, have not been reported before and are characterized here in detail.

## Crucial residues influencing PHL7 stability
By solving the X-ray structures of selected enzyme variants and analyzing their dynamics in MD simulations, we uncovered several structure-function relationships that can inform the engineering of PHL7 and related PET hydrolases. The crucial role of residues 154 and 155, next to the so-called "wobbling" tryptophan (W156), can be explained by their engagement in a buried water network and direct interactions with the catalytic D177 and H209, impacting the geometry

of the catalytic triad. The mutation G155P has a stabilizing effect on PHL7, as demonstrated by a 2.5 °C $T_m$ increase for PHL7-G155P (Supplementary Fig. 16) and a >7.5 °C $T_m$ decrease when removing this mutation from R2M2 (Fig. 6).

Other notable mutations showing consistently stabilizing effects in all variants include Q95G/Y, reported before to impart stability to ICCG[12] and PHL7[28], and Q175E, which was engineered based on a salt bridge between analogous sites to E204/K233 in FAST-PETase[16]. Inspection of the X-ray structures of R4M6 and R4M10 confirms that the designed salt bridge between E175 and R205 is indeed formed in those enzymes (Fig. 8b). Another important salt bridge was engineered through the introduction of mutation E148K, which added the largest contribution to the energetic stabilization of R2M2 (−7.3 kcal mol⁻¹ Rosetta energy). This salt bridge was found to persist in MD simulations of R2M2 and R4 variants (Supplementary Fig. 7). Interestingly, interchanging the residues of the salt bridge K148/D233 kept the $T_m$ constant but diminished activity, as seen by the reduced PET hydrolysis of R4M12 with the salt bridge E148/K233 compared to the R4M6 variant with the non-interchanged salt bridge (Fig. 7c). Hence, the combination K148/D233 is considered most preferable.

## Crucial residues influencing PHL7 activity
An important goal achieved in the current study represents the diminishment of the high buffer dependence of PHL7. Comparison of the electrostatic potential surfaces of PHL7 and R2M2 (Fig. 4d) coupled to site-directed mutagenesis (Fig. 4e) revealed that four charge-changing mutations contribute to the higher activity of R2M2 in 0.1 M

phosphate buffer. The four mutations (Q6E, E148K, D196S, D198P) render the protein surface more neutral and decrease the protein net charge from ca. −6 to −4 at pH 8.0. Furthermore, variants containing only some or none of the four mutations had significantly lower activity in 0.1 M buffer (up to 7-fold reduction), whereas the activity was unchanged compared to R2M2 at 1.0 M buffer conditions (Fig. 4e). Our results are in good agreement with previous observations on halophilic proteins[37,47,48]. Like PHL7, they are coated on their surfaces with a high number of Asp and Glu residues. Halophilic proteins require a highly negative surface charge to remain soluble at high salt concentrations, yet at low ionic strength, this charge density can induce electrostatic repulsion and destabilize the protein fold.

In addition to improving stability, the changed surface electrostatics of R2M2 may also affect the interaction between the enzyme and the PET surface. As shown previously, PET films degraded by PHL7 and other PET hydrolases revealed differences in surface erosion, indicating specific interaction characteristics and degradation pattern[49]. A non-uniform PET surface erosion with the formation of craters was observed for PHL7. The authors attributed this special phenomenon to the negatively charged surface patch on PHL7. According to the proposed mechanistic model illustrated in Thomsen et al.[49], the enzyme position on the PET surface is aligned by repulsion forces between the protein's negatively charged surface and the negative charges of the carboxylic end-groups of the hydrolyzed PET chains. After initial surface attachment, PHL7 would degrade the PET film in one direction, thus leading to the observed non-uniform erosion profile. This model is supported by MD simulation experiments results, which showed that the negatively charged distal side of PHL7 was mostly repelled from the PET surface and that uncharged polar and apolar amino acids dominate instead the interaction of PHL7 with PET[50]. Our engineered PHL7 variants could be useful tools to further validate the proposed model by comparing their adsorption and desorption kinetics in relation to PHL7. Furthermore, the finding that adsorption and desorption processes are linked to the enzymes's PET degradation behavior has important implications for the further engineering of PET hydrolases. Several studies have already focused on modifying the enzyme's surface properties through charge or hydrophobicity adjustments. For example, Nakamura et al. demonstrated accelerated PET adsorption kinetics and higher activity for the enzyme PET2 after introducing positively charged Arg and Lys residues to the protein surface[51]. Furthermore, a similar strategy was followed by Bhattacharya et al., involving the generation of a super-charged variant of the enzyme ICCG[52]. This strategy proved also successful, yielding an ICCG variant with higher thermostability and 2-fold higher efficiency[52].

## Performance of the PHL7 variants under PET recycling conditions

A weakness of many PET hydrolases is a decline of their activity during extended reactions, which is necessary for near-complete substrate conversion in industrial processes. Therefore, we evaluated the long-term stability of the engineered PHL7 variants. A remarkably high stability was observed for R4M9, maintaining its full activity over 1 week at 65 °C, followed by R4M6 with a residual activity of 75% (Fig. 7b). Both variants outperformed ICCG, which had only 63% residual activity after the same period. Likewise, the variants R4M10, R4M12, and R4M15 also displayed significant stability with about 40% residual activity over 1 week at 65 °C. The exceptional durability of R4M9 highlights its potential as a template for designing PET hydrolases suitable for continuous recycling processes with enzyme recovery.

Based on the results of the PET degradation experiments in a bioreactor using high substrate concentrations (Fig. 7c), R4M6, R4M9, and R4M10 were considered as the best performing PHL7 variants. R4M10 achieved 78.7% PET hydrolysis within 24 h, followed by R4M6 (75.7%) and R4M9 (74.4%). All R4 variants exceeded the performance of ICCG, which degraded 51.2% PET within 24 h of reaction. The excellent performance of R4M10 may be the result of the enzyme's high affinity (low $^{inv}K_M$) for PET and a higher binding pocket flexibility observed in the MD simulations compared to the other variants. Interestingly, R4M6 showed the highest degradation rate during the first 6 h of the reaction, consistent with its higher $^{inv}V_{max}$ value (Fig. 7c, d). However, it lost its activity faster than R4M9 and R4M10 during the reaction time, so that its overall performance was not superior to R4M10. The decline of the degradation rates observed over a 48 h reaction in a bioreactor could have been due to enzyme inhibition caused by the accumulation of hydrolysis products, as shown before[53].

Another key determinant of the industrial applicability of PET hydrolases is their activity on highly crystalline PET, as the majority of PET waste−particularly PET bottles−exhibits crystallinity levels above 25%. Previous studies have demonstrated that the catalytic performance of PHL7 and other hydrolases declines sharply when substrate crystallinity exceeds approximately 20%[49]. Accordingly, a critical future objective is the development of PHL7 variants with improved tolerance to high crystallinity, facilitating more efficient degradation of industrially relevant PET materials.

## Comparison with other PHL7 engineering studies

The engineered R4 variants contain many novel mutations, which, to the best of our knowledge, have not been described in the context of PHL7 before. Some other PHL7 variants were recently reported by Groseclose et al.[29], who applied a directed evolution and screening workflow, and by Wang et al.[30], who designed the PHL7 variant Flash-PETase with more than twice the PET-hydrolytic activity of PHL7 in 1.0 M phosphate buffer at 70 °C. However, in both cases, the obtained variants were not designed to perform under low-salt buffer conditions and therefore maintained a strong buffer dependence. In contrast, the tolerance to low-salt buffers endows the R4 variants with a clear advantage for transitioning biocatalytic PET depolymerization processes to large-scale technological applications. Comparison of variants R4M6, R4M9, and R4M10 to the PHL7 variants described by Groseclose et al.[29] reveals Q95Y and Q175E as two common mutations. Both mutations have established beneficial effects, which have been reported in other engineered PET hydrolases, e.g., ICCG[12] and FAST-PETase[16]. Interestingly, one mutation (R111H) in the variant PHL7-Tusas[29] maps to the same position as the R111T mutation in the PHL7 R4 variants. It is possible that this mutation was introduced during the directed evolution process, because it alleviates electrostatic repulsion with the nearby R117, as seen in the R2M2 crystal structure (Fig. 4g). Compared to FlashPETase[30], the R4 variants share E148K as a common mutation. This mutation was identified by Wang et al.[30] due to its large contribution to the energetic stability of PHL7, which is in agreement with our Rosetta energy data (Fig. 3). Interestingly, another mutation in the R4 variants (T145P) was selected for its activity-enhancing effect but was not included in the final FlashPETase[30].

In conclusion, the PHL7 R4 variants developed in this study contain a unique set of mutations, which endow them with high stability and activity under low-salt buffer and high substrate concentration conditions. In particular, R4M6, R4M9, and R4M10 represent promising candidates for a deployment in biocatalytic PET recycling processes. Moreover, our findings provide a template for the further engineering of PET hydrolases, which will make large-scale enzymatic PET depolymerization more efficient, leading to reduced overall process costs.

## Methods

### Site directed mutagenesis and gene synthesis

Site-directed mutagenesis was carried out using the Platinum™ SuperFi II DNA Polymerase (Thermo Fisher Scientific, Invitrogen™). The specific primers and all plasmid sequences used are provided in the Supplementary Information (Supplementary Tables 2 and 3).

Following PCR, DpnI treatment was employed to remove the parental plasmid. This involved incubating 10 μL of the PCR product with 0.4 μL of DpnI (New England Biolabs GmbH, Germany) at 37 °C for 90 min, followed by 80 °C for 20 min. The PCR products were then transformed into DH5α *E. coli* cells (New England Biolabs GmbH, Germany), which were subsequently plated on agar plates containing kanamycin (30 μg mL$^{-1}$). Positive colonies underwent plasmid extraction using the Monarch Plasmid Miniprep Kit (New England Biolabs GmbH, Germany). Complete genes of R1 and R2 mutants were obtained from Genscript in pET26b(+) plasmid. The sequences of all mutants and synthetic genes were confirmed by Sanger sequencing.

## Protein purification

*E. coli* BL21 cultures were cultivated at 37 °C and 220 rpm in either LB or TB medium, supplemented with 30 μg mL$^{-1}$ kanamycin, until they reached an optical density at 600 nm (OD$_{600}$) of 0.6. Subsequently, the temperature was lowered to 18 °C, and protein expression was induced by adding isopropyl-β-D-1-thiogalactopyranoside (IPTG) to a final concentration of 0.1 mM. Cultures were further incubated at 18 °C for 16 h with gentle agitation at 120 rpm. Cells were then harvested and disrupted in buffer A (50 mM sodium phosphate, pH 7.4, 200 mM NaCl) by sonication. The lysate was clarified by centrifugation at 14,000 × *g* for 30 min, and the resulting supernatant underwent sterile filtration before being subjected to affinity purification using an ÄKTA purifier system (Cytiva, Marlborough, MA, USA). The His-tagged proteins were purified using a 5 mL HisTrap FF column (Cytiva, Marlborough, MA, USA) equilibrated in buffer A, and were eluted using the same buffer containing 250 mM imidazole. Hereafter, the protein samples were subjected to size exclusion chromatography (SEC) using a HiLoad 16/600 Superdex 75 pg column (Cytiva, Marlborough, MA, USA) equilibrated in buffer A. The purity of the enzymes was confirmed by SDS PAGE using 15% polyacrylamide gels (Tris-glycine buffer). A prestained molecular marker (PS10 PLUS;11–180 kDa, Gene On, Ludwigshafen, Germany) was used as a protein ladder for molecular weight evaluation. Staining was performed with Coomassie Brilliant Blue G-250 (Sigma-Aldrich, San Luis, MO, USA).

High-yield expression of variants R4M6, R4M9, R4M10, and R4M12 was achieved using the enGenes-X-press system for growth-decoupled recombinant protein expression, as described by Stargardt et al.[54]. In summary, enGenes-X-press cells, a modified *E. coli* BL21 strain encoding a bacteriophage T7 derived RNAP inhibitor Gp2[54], were cultivated in glycerol semi-synthetic medium (G-SSM) supplemented with 30 μg mL$^{-1}$ of kanamycin at 37 °C and 220 rpm until an OD$_{600}$ of 8–9 was reached. At this point, the temperature was reduced to 25 °C, and expression was induced by the addition of IPTG and L-arabinose to attain final concentrations of 1.0 mM and 10 mM, respectively. Following expression for 16–18 h at 25 °C and 220 rpm, the cells were harvested by centrifugation at 9000 × *g* for 30 min and the cell culture medium was loaded onto a 5 mL HisTrap FF crude column (Cytiva, Marlborough, MA, USA). The recombinant proteins were eluted with buffer A containing 250 mM imidazole. The fractions containing the respective recombinant proteins were collected and applied to SEC, using a HiLoad 16/600 Superdex 75 pg column (Cytiva, Marlborough, MA, USA) equilibrated in buffer A. Sample purity and homogeneity were confirmed by SEC and SDS PAGE. Chromatograms and SDS gels from purifications of R4M6, R4M9, R4M10, R4M12, and ICCG are displayed in Supplementary Fig. 18.

## PCL plate clearing assays

To test if mutants were active, preliminary assessments were conducted using polycaprolactone (PCL) plate clearing assays to evaluate the extent of PCL degradation as described before[55]. The presence of a zone of clearance around colonies served as an indicator of PCL hydrolyzing activity. To create PCL agar plates, a suspension of PCL nanoparticles was introduced to autoclaved LB agar (6% (v/v)) under continuous stirring at 60 °C. Subsequently, the mixture was supplemented with 0.5 mM IPTG and 37 μg mL$^{-1}$ kanamycin before being poured into Petri dishes. After the mixture had cooled and solidified, the Petri dishes were then inoculated with recombinant *E. coli* cells harboring the respective enzyme plasmids, and they were incubated at 37 °C overnight.

## PET weight loss assays

The hydrolytic activity was first assessed by gravimetric assays, and afterward, the degradation rates were determined by impedance spectroscopy[45]. For gravimetric assays, amorphous PET films of 3 cm × 0.5 cm (about 45 mg) (3–5% crystallinity) (Goodfellow, Hamburg, Germany) were washed with 0.5% (w/v) SDS, deionized water, and 100% ethanol and dried overnight at 50 °C. Then the films were weighed, and placed in reaction vials with the purified enzymes (0.55 mg enzyme per gram PET) in a total volume of 1.8 mL in 0.1 or 1 M K$_2$HPO$_4$-NaOH, pH 8.0. The incubation was carried out for 12, 14, or 16 h, depending on how active the variants were. The activity of the variants was normalized to the activity of the WT enzyme at the corresponding times. The PET films were incubated under vigorous shaking at 65 and 70 °C. Following the incubation, the films underwent washing, drying, and gravimetric analysis to determine weight loss.

## Melting point determination

Nano-differential scanning fluorimetry (nanoDSF) was utilized to determine the apparent melting temperature (T$_m$) of PHL7 mutants using a Prometheus NT.48. This method involves tracking changes in fluorescence signals emitted by the protein's tryptophan residues at specific wavelengths, namely 330 and 350 nm, and linking these alterations to shifts in temperature[56]. Analysis was conducted on purified enzyme fractions in 50 mM Na$_2$HPO$_4$, pH 7.4, 200 mM NaCl buffer at a concentration of 0.15 mg mL$^{-1}$. The temperature was ramped from 20 to 95 °C at a rate of 1 °C per minute. The presented data represents mean values ± standard deviation for a sample size of $n = 3$.

## Impedimetric determination of PET degradation rates

PET film degradation rates of PHL7 mutants were also determined using a method based on impedance spectroscopy, as described previously[45]. This method observes changes in capacitive resistance exhibited by polymer films, functioning as barriers within paired reaction chambers, each housing a platinum electrode. Each chamber was divided by a square PET film (44.7 mm$^2$, ~225 μm thickness) (3–5% crystallinity) and contained either 1 M or 0.1 M K$_2$HPO$_4$ buffer (pH 7.8) along with 1.18 μM of enzyme (30 μg mL$^{-1}$). Reactions were carried out at 65 °C or 70 °C for 20 h. Impedance measurements were recorded every 3 min across a frequency range of 1000 Hz to 1 MHz using an ISX-3v2 high-precision impedance analyzer (Sciospec Scientific Instruments GmbH, Germany), applying a 100 mV signal amplitude with 51 frequency points. Data acquisition and automation were managed by custom software (IMATadvanced, 2021). Raw impedance data were processed via custom software (Universal EC-Fitter, 2024) using a simplified Randles equivalent circuit model to derive capacitance values. These values were then converted into PET thickness changes, assuming a relative permittivity (ε$_r$) of 3.3 and an initial mechanically verified thickness of ~225 μm. The enzymatic degradation rates were determined from the change in layer thickness per hour. The mean of the rates between 3 and 8 h was plotted as the mean rate.

## Inverse Michaelis-Menten kinetics

PET films (18 mg, corresponding to a PET load of 10 g L$^{-1}$) were prepared as previously described[46]. Reactions were conducted in 1.8 mL tubes containing 1 M K$_2$HPO$_4$ buffer (pH 8.0) and varying enzyme concentrations (0–0.6 μM). The tubes were incubated for 4 h at 65 °C in an orbital incubator (Eppendorf, Hamburg, Germany) with vigorous shaking. Reaction times were chosen to lie within the linear range of

 

enzymatic activity. All reactions were performed in duplicate. After incubation, the supernatants were transferred to UV-transparent microplates (Corning, Wiesbaden, Germany), and the release of terephthalic acid equivalents (TPA eq.) was quantified by measuring absorbance at 240 nm using a microplate reader (Agilent Technologies, Waldbronn, Germany). Quantification was based on a standard calibration curve, which was generated by preparing a series of TPA dilutions ranging from 0 to 1 mM in a final volume of 100 μL. The absorbance of each dilution was measured at 240 nm. The resulting data were fitted to the equation $v = {}^{inv}V_{max} \cdot [E] / {}^{inv}K_M + [E]$, as described previously[46], using GraphPad Prism.

## PET degradation in a pH-controlled bioreactor

Degradation experiments using high substrate concentrations were conducted in a cylindrical double-walled bioreactor. A total of 50 g of low-crystalline PET (post-consumer clamshell packaging of various brands) was cut into approximately $1 \times 1$ cm pieces. The crystallinity of these PET containers was previously determined to be between 3 and 5%[26]. The PET flakes were suspended in 500 mL of 0.1 M $K_2HPO_4$ buffer (pH 8), pre-heated to 65 °C using a thermostatic circulator. The reaction was initiated by adding 40 mg of purified enzyme, and reactions were tracked for 48 h. Agitation was maintained with a paddle stirrer at 220 rpm. The pH was kept constant at 8.0 by automatic titration using a peristaltic pump supplying 5 M NaOH. The NaOH consumption was continuously monitored gravimetrically. The theoretical amount of hydrolyzed PET was calculated based on the NaOH consumption during the reaction. It was assumed that the enzymatic depolymerization of PET yields TPA as the primary acidic product, which is neutralized by NaOH in a 1:2 molar ratio (1 mol TPA requires 2 mol NaOH). Accordingly, the moles of TPA formed were calculated as $0.5 \times$ mol NaOH consumed. Based on the molecular weight of the repeating unit of PET (192.2 g mol⁻¹), the mass of theoretically depolymerized PET was determined.

## Rosetta computational design

Multi point mutants of PHL7 were designed using the Rosetta PROSS algorithm[31] through the PROSS2 webserver[57]. The PROSS method suggested seven designs, comprising an increasing number of mutations relative to one another, and two additional designs, comprising a larger number of more permissive mutations. Design of the catalytic triad residues (S131, D177, H209) and of first shell residues around the catalytic triad was disabled. Mutants were analyzed in terms of their total Rosetta energy and per-residue energy using the ENDURE webserver[58] and the Rosetta energy function 2015 (ref2015)[59].

## Molecular dynamics simulations

MD simulations were performed with Gromacs 2023[60] patched with Plumed 2.9[61]. Proteins were modeled with the Amber99sb-ildn force field[62,63]. For MHET and 4xMHET, we used the Sage force field[64] and AM1-BCC charges. Dihedrals were then refitted with AIMNet2 using PlayMolecule (http://www.playmolecule.org/)[65]. Additional protein-ligand interaction corrections were introduced to account for specific interaction features of oxyanion holes (Supplementary methods)[66]. For all systems, we used a multi-step equilibration scheme described before[67].

For the first experiment, we modeled WT, R2M2, R4M6, and R4M10 with and without substrate (MHET) at 300 and 343 K, resulting in a total of 16 systems. For each system, we performed equilibration followed by 100 ns production in 5 independent replicates. For HREX simulations done for WT, R4M6, R4M9, and R4M10 with 4xMHET, we used a set up similar to previous work[68], with additions described in Supplementary methods. We ran HREX for 500 ns per lambda window, and discarded the first 100 ns when analyzing the results. For free forms of the latter set of systems, we performed OPES enhanced sampling simulations[69] to explore the behavior of the binding site

(Supplementary methods). These simulations were run for 100 ns in 3 replicates.

## Protein crystallization

Crystallization of R2M2, R2M2-P155G, R2M2-A186S, and R2M2-P155G/A186S was prepared by transferring the protein into a buffer consisting of 10 mM Tris (pH 7.4) and 10 mM NaCl as described previously[27]. In contrast, R4M6, R4M10, and R4M12 were maintained in their original buffer containing 50 mM sodium phosphate (pH 7.4) and 200 mM NaCl. All samples were concentrated to 8–10 mg mL⁻¹ prior to crystallization.

Initial crystallization trials for R2M2, R4M10, and R4M12 were performed using a high-throughput screening approach, testing approximately 400 conditions in a 96-well plate format. Crystallization drops were prepared using the sitting-drop vapor diffusion method, with 200 nL drops consisting of equal volumes of protein solution and reservoir solution (100 nL each), equilibrated against 100 μL reservoir solution. A Mosquito Xtal3 pipetting robot (SPT Labtech, Melbourn, England) was used to set up these drops. Conditions yielding initial crystals were subsequently selected and manually reproduced using the hanging-drop method, where 1 μL of protein solution was mixed with an equal volume of reservoir solution. Additional optimization was required for R4M12 to obtain high-quality crystals.

Crystals of R2M2 formed within 1 week under conditions containing 0.1 M sodium citrate (pH 4.0), 10% (w/v) PEG 6000, and 3% (v/v) 1,6-hexanediol. This crystallization condition was also suitable for R2M2-P155G, R2M2-A186S, and R2M2-P155G/A186S variants. Crystals of R4M6 and R4M10 were obtained using a reservoir solution containing 1.1 M sodium malonate, 0.1 M HEPES (pH 7.0), and 0.5 % (v/v) Jeffamine ED-2003. R4M12 crystallized under conditions comprising 0.1 M sodium citrate (pH 5.6), 20% PEG 4000, and 20% (v/v) isopropanol. All crystals grew within 1 week at 19 °C. Single crystals measuring 0.5–1 mm in length were flash-frozen in liquid nitrogen using a cryobuffer consisting of the crystallization buffer supplemented with 15% PEG400.

## X-ray structure determination

Crystal diffraction data were collected using synchrotron radiation at beamlines P13 or P14 of PETRA III (DESY Synchrotron, Hamburg, Germany). The diffraction data were indexed, integrated, and scaled using XDS (version 10 January 2022 and 30 June 2023)[70] and STARANISO (version 2.3.74) as implemented in ISPyB[71] at DESY.

Structures were determined using molecular replacement (MR) with Phaser within CCP4i2 (CCP4 version 8.0)[72] or by rigid-body refinement with REFMAC (version 5.8.0425)[73]. Refinement was performed using jelly-body refinement in REFMAC (version 5.8.0425), and model building was completed in Coot (version 0.9.8.93 and 0.9.8.95)[74]. Final refinements were conducted in Phenix (version 1.20.1_4487 and 1.21.2_5419)[75,76].

The structure of R2M2 was determined by molecular replacement using the known structure of PHL7 (PDB ID: 7NEI)[26] as a search model. The refined R2M2 structure was then used as a starting model for rigid-body refinement to determine the structures of R2M2-P155G, R2M2-A186S, and R2M2-P155G/A186S, and as a search model for molecular replacement to determine the R4M6 structure. The R4M6 structure was subsequently used as a starting model for determination of the structures of R4M10 and R4M12 via rigid-body refinement and molecular replacement, respectively. All structures were refined and completed using programs mentioned above. Data collection and refinement statistics are listed in Supplementary table 4.

## Reporting summary

Further information on research design is available in the Nature Portfolio Reporting Summary linked to this article.

## Data availability

The coordinates and structure factors of PHL7-R2M2, R2M2-P155G, R2M2-A186S, R2M2-P155G/A186S, R4M6, R4M10, and R4M12 have been deposited in the Protein Data Bank under accession codes 9QNM, 9QT8, 9QV8, 9QVA, 9QYA, 9QYB, and 9QYC, respectively. The customized scripts used for running and analyzing the MD simulations of PHL7 variants are available via a Zenodo repository under https://doi.org/10.5281/zenodo.18650810. All other computational and experimental data generated and analyzed in the study are included in the manuscript, in the supporting material or in the supplementary source data files, and are available from the corresponding author(s) upon request. Source data are provided with this paper.

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

## Acknowledgements

W.Z., C.S., P.B.S., and Z.Z. acknowledge funding from the European Union's Horizon 2020 research and innovation program under grant agreement no. 887913 (ENZYCLE) and from the Sächsische Staatsministerium für Wissenschaft, Kultur und Tourismus (SMWK) (project no. 100387903) in the framework of the ERA CoBioTech project MIPLACE. G.K., P.B.S., J.G., and N.G. acknowledge funding from the European Union's Just Transition Fund (JTF) InfraProNet programme (project no. 100704504). G.K. acknowledges further financial support by the Federal Ministry of Education and Research of Germany and by Sächsische Staatsministerium für Wissenschaft, Kultur und Tourismus in the programme Center of Excellence for AI-research, Center for Scalable Data Analytics and Artificial Intelligence Dresden/Leipzig, project identification number: ScaDS.AI. Manuscript publication was supported by funding from the Open Access Publishing Funds of Leipzig University supported by the German Research Foundation within the programme Open Access Publication Funding. The authors further thank the high performance computing centers at Leipzig University and at the NHR center of TU Dresden for providing the computational resources.

We further acknowledge DESY (Hamburg, Germany), a member of the Helmholtz Association HGF, and the EMBL for the provision of experimental facilities at synchrotron beamlines P13 and P14. We thank the on-site scientists Kiril Kovalev and Gleb Burenkov, and Dr. Renato Weiße for their support during data collection.

## Author contributions

P.B.S., J.G., and A.S. expressed and purified all enzyme variants, measured protein $T_m$ data, and recorded and analyzed PET weight loss, impedance spectroscopy, and enzyme kinetics data with support from N.G., F.C., Z.Z., A.Z., R.F., and E.B. A.U. and N.S. performed protein crystallography and determined the X-ray structures of all PHL7 variants and prepared figures for the manuscript. A.Z. performed the MD simulations of all PHL7 variants and prepared figures for the manuscript. J.D.Z. and F.E. carried out the Rosetta PROSS calculations and designed the R1 and R2 variants. J.M., W.Z., and N.S. supervised the experiments and acquired funding for the project. G.K. and C.S. conceived the project, designed and supervised the experiments, and acquired funding. C.S. performed the bioreactor experiments of all enzyme variants. G.K., P.B.S., J.G., A.U., and A.Z. prepared the figures for the paper and wrote the manuscript with input from all authors.

## Funding

## Competing interests

The authors G.K. and C.S. of Leipzig University declare that they have filed a patent application related to the enzyme variants described in this study. The remaining authors declare no competing interests.

## Additional information

[1]Institute for Drug Discovery, Leipzig University, Leipzig, Germany. [2]Institute of Bioanalytical Chemistry, Centre for Biotechnology and Biomedicine, Leipzig University, Leipzig, Germany. [3]Interdisciplinary Center for Bioinformatics, Leipzig University, Leipzig, Germany. [4]Centre for Biotechnology and Biomedicine, Biochemical Cell Technology, Leipzig University, Leipzig, Germany. [5]Institute of Analytical Chemistry, Leipzig University, Leipzig, Germany. [6]Center for Scalable Data Analytics and Artificial Intelligence, Leipzig University, Leipzig, Germany. [7]These authors contributed equally: Paula Blázquez-Sánchez, Jonas Gunkel, Abibe Useini. ✉e-mail: christian.sonnendecker@uni-leipzig.de; georg.kuenze@uni-leipzig.de

