## [Transparent Peer Review file · Nature Communications]

Computational engineering of the polyester hydrolase PHL7 for efficient poly(ethylene terephthalate) degradation in biocatalytic recycling processes

Corresponding Author: Dr Georg Künze

Version 0:

Reviewer comments:

Reviewer #1

(Remarks to the Author)

This manuscript describes the engineering of Polyester Hydrolase Leipzig 7 (PHL7) to improve its stability and activity for PET degradation. The authors employed computational design (Rosetta PROSS) and rational mutagenesis to introduce up to 24 mutations, generating variants such as R4M6, R4M9, and R4M10 with apparent melting temperatures of 88–95 °C and enhanced activity. Mechanistic investigations integrating X-ray crystallography and molecular dynamics (MD) simulations highlight the roles of salt bridges, surface charge redistribution, active-site water networks, and binding-pocket flexibility. The dataset is extensive (>60 mutants, seven crystal structures, and large-scale MD simulations). However, the manuscript suffers from significant issues in conceptual framing, methodological rigor, and data presentation. The logical flow is difficult to follow for non-specialists, and more importantly, the study does not convincingly articulate how the engineered PHL7 variants advance theoretical understanding in enzymatic PET recycling. Additionally, some assay conditions deviate from field standards, raising concerns about the reliability of the conclusions, which currently limit its suitability for Nature Communications.

Major Concerns

1. Numerous PET hydrolases surpassing ICCG have been reported in the past two years (e.g., Kubu-PM12 (ref 25), ICCG-A2 (ref 21), TurboPETase (ref 24)). The current manuscript does not benchmark against these enzymes, nor do the reported performance metrics demonstrate clear novelty. The mechanistic explanations—salt bridges, surface charge, water networks, binding-pocket flexibility—remain conventional in enzyme engineering, without offering transformative design principles. To elevate its impact, the study must present a cohesive theoretical framework that generalizes beyond incremental observations.

2. Industrial benchmark conditions are inappropriate

The section “The R4 variants of PHL7 outperform ICCG and HotPETase in PET degradation under industrial process conditions” is problematic.

First, HotPETase should not be used as a benchmark, since multiple studies have already demonstrated that its performance under true industrial conditions is not superior (e.g. ACS Catal. 2023, 13, 13156–13166). More appropriate benchmarks would be ICCG-A2, TurboPETase, and Kubu-PM12, all of which have been tested under authentic industrial settings and shown to outperform ICCG.

Second, the so-called “industrial conditions” employed in this study do not actually reflect standard industrial parameters. For example, the reactions for ICCG degradation (ref 12) were carried out in aqueous environments rather than in 0.1 M buffer solutions, and the substrate concentration was above 20% (w/w) PET powders, not the 10% (w/w) PET flakes used here. The use of PET flakes is particularly concerning. Some PETases exhibit higher degradation performance toward PET films compared to PET powders, likely due to differences in polymer surface adsorption and binding properties. However, all industrial protocols preferably use PET powders rather than PET films, as the increased surface area accelerates degradation and reduces processing time. Indeed, PET powders could be fully degraded within 24 hours by ICCG (for more details, please refer to ref 21 and ref 24 and ACS Catal. 2023, 13, 13156–13166). By contrast, the results presented in Figure 7C of this manuscript do not align with the previously reported industrial-level performance of ICCG. This discrepancy may be attributable to the use of PET films instead of powders, as well as the fact that enzyme concentrations may not have reached saturation. Therefore, I strongly recommend that the authors repeat these experiments under standardized industrial

conditions—namely, using appropriate buffer systems, PET powder substrates at $\geq 20\%$ (w/w), sufficient enzyme loadings to reach saturation, and benchmarks that reflect state-of-the-art enzymes (A2, TurboPETase, Kubu-PM12). Without such standardized testing, the current dataset cannot provide reliable or comparable activity data for the field.

3. The manuscript emphasizes the impact of phosphate buffer concentration on enzyme stability, noting that PHL7 becomes unstable under 0.1 M phosphate (line 80). This suggests that the stability of these mutants may be significantly influenced by the solvent environment. However, the thermal stability (apparent melting temperature, T_m) of mutants is measured in a uniform buffer (50 mM Na_2HPO_4 pH 7.4, 200 mM NaCl) used for protein purification (line 863, line 807). This raises a critical concern: the T_m values obtained under these conditions may not reliably reflect stability in reaction environments, such as 1 M KH_2PO_4 buffer, or 0.1 M KH_2PO_4 buffer (without NaCl). The authors should directly compare T_m values of key variants (e.g., PHL7, and R4M6/R4M9/R4M10) in buffers mimicking actual reaction conditions. This would clarify whether the observed stability trends hold across different ionic strengths and salt compositions.

4. While the authors report $>90\%$ PET degradation for PHL7-R4 variants based on sodium hydroxide consumption in bioreactor experiments, this single metric may not fully capture the degradation process. In contrast, reference 12 employs multiple complementary metrics for ICCG—including residual plastic dry weight, terephthalic acid (TPA) yield, and ethylene glycol (EG) yield—to robustly validate degradation efficiency. To strengthen the credibility of their findings, the authors should incorporate at least one additional quantitative measure (e.g., residual plastic dry weight or direct quantification of TPA/EG) alongside NaOH consumption.

5. Unclear presentation of computational pipeline

In the section "Rational design of PHL7 identifies activity enhancing mutations," the presented mutations are described as the final step of rational design rather than serving as starting points for subsequent computational iterations. This arrangement obscures the sequential logic of the engineering process, making it difficult for readers to track how each mutant evolves through successive optimization stages. The current depiction of mutant relationships in Figure 3a uses directional arrows that are prone to misinterpretation as iterative optimization. In reality, these variants were computationally generated and experimentally tested in parallel, without an experimentally validated stepwise optimization relationship. The multi-colored circular schematic in Figure 2a overwhelms readers with excessive visual complexity, hindering the extraction of key information. Overall, the narrative of computational design remains inaccessible to readers, undermining the technical credibility of the manuscript. The authors must restructure the workflow description and associated visuals to present a transparent, logically coherent engineering pipeline that aligns with standard conventions in computational enzyme design studies.

In summary, while this manuscript presents a substantial dataset, including >60 engineered mutants, seven high-resolution structures, and MD simulations, its scientific contribution is limited by insufficient benchmarking against state-of-the-art enzymes, use of non-standard industrial conditions, and lack of theoretical advancement.

Reviewer #2

(Remarks to the Author)

See the attached file.

Reviewer #3

(Remarks to the Author)

This manuscript by Blazquez-Sanchez et al. presents a study on the engineering of a highly active derivative of a polyethylene terephthalate (PET) hydrolase that is thermostable in relatively low ionic strength. This hydrolase is being investigated for the industrial-scale degradation of plastics based on PET, but its dependence on high ionic strength for stability minimizes its use for industrial-scale bioreactor applications. Thus, the authors used a combination of *in silico* protein modelling, protein engineering, biochemical characterization and x-ray crystallography to generate an enzyme with the desired properties. The research represents a tour de force in its comprehensive design and performance, such that the underlying mechanisms driving the activity and thermostability could be discerned. This, and the importance of the research that addresses a major ecological concern involving plastic waste pollution, impacts the significance of the work.

This reviewer has only two issues the reviewers may want to address. The first concerns the last paragraph of the Introduction which, in effect, is a reiteration of the Abstract. This could be replaced with a simple statement of purpose, and Figure 1 could be integrated into the Results section, perhaps involving an introductory statement regarding experimental design.

The second, and also a relatively minor issue, pertains to the experimental design of the thermostability analysis (Melting Point Determination). As a major focus of this research was to generate a hydrolase with enhanced stability in low ionic strength (0.1 M phosphate buffer), compared to its current property of requiring 1 M concentrations, it is curious that the thermostability assays were conducted in neither conditions, but rather something more in between (*viz.* 50 mM phosphate with 200 mM NaCl).

Reviewer #4

(Remarks to the Author)

This work follows on from the discovery and characterization of a thermotolerant PETase, PHL7, that was isolated from a compost metagenome. While this enzyme showed promise in terms of its stability and activity profiles relative to some benchmark enzymes currently used for PET deconstruction, this particular enzyme required unusually high salt concentrations, up to 1 M phosphate, to achieve this. As noted by the authors, this represents a significant limitation for the

use of this enzyme in an industrial setting where buffer and salt requirements should be minimized, as demonstrated in multiple TEA and LCA studies. However, the advantage of enzymes that can maintain high activity at elevated temperatures, ~65 deg C, and even remain active over multiple cycles, remains a key target for the field.

In this study, the native enzyme is engineered to produce a performant variant that retains/surpasses the activity profile, increases the thermotolerance, and mitigates the requirement for high salt conditions. In fact, and a rather a hidden outcome of this work up front, is that one of their variants retained full activity after a week at 65 deg C (see comment below). This is a well conducted study that leverages significant expertise in the team in terms of protein engineering and analysis. The combination of modelling, X-ray crystallography, and stability/activity measurements is well executed and presented.

Specific comments:

While most PETase enzymes require some salt to maintain their optimum activity, PLH7 is rather an outlier in terms of the extreme requirement for molar concentrations. In the introduction, high salt requirement is framed as a general issue. To be more transparent to the reader, it would be better to note that several benchmark enzymes (as described), have more modest salt requirements. Published Kubu-P bioreactor experiments were performed at 100 mM Na₂HPO₄ for example. Thus, the focus here is on reducing the unusually high salt reliance specifically for PLH7 in order to bring it closer in alignment to the industrially relevant benchmarks, including LCC variants and Kubu-P, which is indeed the target and successful outcome here. This remains a valid goal, but rewording will helpfully delineate this study from engineering this parameter in PETases more generally.

The strategy is well considered and nicely depicted in Figure 1, making a complex and involved process generally accessible. However, I suggest the each of the sections that describe the 4 rounds are shortened for further readability. Some of this text could be productively moved to the SI, allowing the main text to focus on more concise strategies and outcome for each. Listing every mutation and result in the text is distracting from the main messages, particularly when much of these data are shown in the figures. Examples, lines 422-435, lines 473-509, where the text is rather dense.

It would be interesting to measure the actual T_m of variant R2M2 with a compatible method (line 240), such as DSC. Without these data, this should not be plotted as a single data point in Fig 3a. Similarly, the delta T_m for this variant is unknown and may be higher (Fig 3b).

The significance of the stability of the R4M9 variant is somewhat understated. Continuous processing and/or enzyme recovery are limited by long-term stability, but have great potential to positively transform the recycling process. While out of the scope experimentally of this study, it would be interesting to see the longevity limits here, towards the goal of months for example. A comment in the conclusions would be welcome, including the potential translation of these properties to other enzymes with similar scaffolds.

Please report the crystallinity of all the substrates used in the main text: the semi-crystalline PET films (line 531), the low-crystalline PET flake (line 584), and low-crystalline clamshell (line 901). In the industrial context that is presented, a comment should be made on the performance of these enzymes on model substrates with defined high crystallinity, as compared to the amorphous films that was tested.

The reverse experiment for the charged mutations is well considered and insightful. An SI figure (and main text commentary) plotting the pI of the variants, and including other known proteins, would be helpful for context. (around lines 275-286)

The lack of major structural rearrangements, coupled with the subtle changes observed by MD are intriguing, given their significant biophysical effects on activity. While the structural comparisons (including in the SI) are comprehensive between the parents and engineered variants, it would be valuable to have a comparison between at least the wt and an R2 or R4 variant with LCC_ICCG or similar as a better segway to the activity data that follows. (or alternative in section beginning line 608). It would be interesting to see if there is a trajectory towards higher similarity with some of the benchmark enzymes over the rounds.

In the conclusions, it would be interesting to reflect on the reasons behind the native enzyme's high phosphate requirements, and link to the observation that this may correlate with the acid patch on the wt enzyme.

The commentary around the mechanistic model for PET surface binding and interfacial catalysis could be more easily described with reference to an SI figure depicting the likely position of the PET surface in relation to the modified charged amino acids.

This work is well conducted and provides novel and valuable insights to this growing community. I recommend publication in Nature Comms with these minor corrections.

Version 1:

Reviewer comments:

Reviewer #1

(Remarks to the Author)

The revised manuscript has addressed most concerns from the first review, particularly by improving experimental validation, data presentation, and benchmarking. The added comparisons against recently reported enzymes (e.g., TurboPETase, LCC-A2) and the use of multiple degradation metrics (NaOH consumption and residual PET weight) improve transparency and reproducibility. These revisions strengthen the experimental reliability and readability. However, critical limitations remain regarding (i) methodological innovation and (ii) marginal industrial-scale benefit relative to frontier bottlenecks. In its current form, the study is best described as a competent application of established protein design tools to a single enzyme scaffold, with incremental mechanistic rationalizations that do not yield a new engineering principle generalizable beyond the presented case.

The authors frame the work as an “in silico–first, multipoint design methodology” that “substantially reduces experimental testing effort” and argue that the PROSS-based process explores sequence space more broadly than “sequential, greedy optimization schemes”. Yet, the computational component relies on existing, widely used infrastructure (Rosetta/PROSS) and is even executed via the PROSS2 webserver. I did NOT find (a) a new scoring function, (b) a new predictive model/learning strategy, or (c) a new sampling/search algorithm. The field already contains numerous examples where existing computational stabilization/design frameworks (e.g., PROSS/FireProt/FRESCO) are used to reduce experimental burden and enable multipoint mutagenesis—this is an established paradigm rather than a novelty claim.

Recently, several high-impact PETase papers that are “method-driven” make their novelty very explicit—e.g., the VenusMine pipeline integrates language-model embeddings + representation-tree clustering with structure/sequence retrieval and multi-tier screening. process integrates language model embedding and representation tree clustering, combined with structure/sequence retrieval and multi-layer screening (Nat Commun 16, 6211 (2025)); a study published in Science in 2025 developed a “landscape construction and sampling” framework, utilizing neighborhood analysis and hierarchical/cluster sampling to track the fitness landscape (Science 387, eadp5637(2025)). Against such standards, the present manuscript’s computational component does NOT read as a methodological advance. Given this context, the authors should substantially tone down statements implying algorithmic or methodological innovation, and instead clearly position the work as an application/engineering case study.

And more importantly, even after revision, the engineered variants appear to match—but not convincingly exceed—the performance envelope already achieved by leading PET hydrolases reported in the last 1–2 years (e.g., TurboPETase and LCC-A2) under standardized benchmarking. In this context, the manuscript reads primarily as a catch-up study: it arrives ~2 years later at a similar performance level, yet does not deliver either a new design paradigm or a demonstrable process-level advantage. The study still does not engage sufficiently with the frontier bottlenecks emphasized in current PET biorecycling research: degradation of high-crystallinity PET, performance under realistic mixed-waste streams, and process concepts that reduce alkali dependency / downstream neutralization burden (Nat. Chem. Eng. 2025, 2, 309–320). As a result, the contribution appears incremental and non-leading in time, raising a fundamental question of editorial fit: if comparable performance was demonstrated ~2 years ago, what is the distinct, current advance that warrants publication at the same journal tier now? In the absence of a new methodology, new mechanistic principle, or clear process-level breakthrough, I do not see a compelling justification.

Overall, in its current form, the work does not provide a new computational paradigm nor a step-change in PET biorecycling performance under frontier-relevant constraints, and thus I do not yet see a strong justification for Nature Communications.

Reviewer #2

(Remarks to the Author)

The authors have reasonably addressed almost all comments raised by the reviewers and, the current, version has improved with respect to the previous one. In this regard, I recommend the publication of this study in Nature Communications.

Nevertheless, I think that the generation of the R2 variants is still confusing. For instance, I do not understand why the authors have incorporated some mutations from R1M7 to generate R2M1 and R2M2 according to an energetic criterion. From Supplementary Figure 3, I would have selected mutations from R1M1, R2M2 and R1M4 to propose R2M1 and mutations from R1M6 and R1M9 plus R2M1 to generate R2M2. The clarification of this point can be relevant for others groups which can adopt this computational strategy for other enzyme engineering applications.

Reviewer #4

(Remarks to the Author)

The authors have comprehensively addressed my comments and the majority of those similar and additional comments from the other reviewers. The manuscript has benefited from the additional analysis, and the narrative and key messages are now clearer. This is an excellent piece of research that will benefit the field. I recommend publication in Nature Communications without further revision.

Reviewer #1:

This manuscript describes the engineering of Polyester Hydrolase Leipzig 7 (PHL7) to improve its stability and activity for PET degradation. The authors employed computational design (Rosetta PROSS) and rational mutagenesis to introduce up to 24 mutations, generating variants such as R4M6, R4M9, and R4M10 with apparent melting temperatures of 88–95 °C and enhanced activity. Mechanistic investigations integrating X-ray crystallography and molecular dynamics (MD) simulations highlight the roles of salt bridges, surface charge redistribution, active-site water networks, and binding-pocket flexibility. The dataset is extensive (>60 mutants, seven crystal structures, and large-scale MD simulations). However, the manuscript suffers from significant issues in conceptual framing, methodological rigor, and data presentation. The logical flow is difficult to follow for non-specialists, and more importantly, the study does not convincingly articulate how the engineered PHL7 variants advance theoretical understanding in enzymatic PET recycling. Additionally, some assay conditions deviate from field standards, raising concerns about the reliability of the conclusions, which currently limit its suitability for Nature Communications.

We appreciate the reviewer's constructive feedback and thoughtful suggestions for improving the manuscript. As detailed in our point-by-point responses below, we have substantially revised our work to address these concerns. In particular, we have incorporated additional benchmarking against state-of-the-art PET hydrolases, performed further bioreactor experiments under relevant conditions, and refined the description and visualization of our engineering workflow to improve conceptual clarity. We hope that these revisions strengthen both the rigor and accessibility of the manuscript.

1) Numerous PET hydrolases surpassing ICCG have been reported in the past two years (e.g., Kubu-PM12 (ref 25), ICCG-A2 (ref 21), TurboPETase (ref 24)). The current manuscript does not benchmark against these enzymes, nor do the reported performance metrics demonstrate clear novelty. The mechanistic explanations—salt bridges, surface charge, water networks, binding-pocket flexibility—remain conventional in enzyme engineering, without offering transformative design principles. To elevate its impact, the study must present a cohesive theoretical framework that generalizes beyond incremental observations.

We have expanded our benchmarking analysis to include recently reported high-performance PET hydrolases, specifically TurboPETase (Cui 2024, 10.1038/s41467-024-45662-9) and LCC-A2 (Zheng 2024, 10.1021/acscatal.4c00400), and now compare the activities of our PHL7 R4 variants to these enzymes. We have also conducted additional PET hydrolysis experiments under more demanding conditions, including reactions with 20% (w/w) PET, to more comprehensively evaluate catalytic performance and industrial relevance. The results of these experiments are described in our answer to the next comment and shown in the updated Figure 7 in the paper.

Regarding the novelty of our engineering strategy, we wish to emphasize that our approach relies on an *in silico*—first, multipoint design methodology, in which a small number of computationally generated variants carrying numerous mutations are experimentally tested. This substantially reduces experimental testing effort while still achieving high success rates in stability enhancement. This strategy contrasts with the stepwise, combinatorial, or site-saturation—driven engineering approaches commonly applied to PHL7 (Wang 2025, doi.org/10.1016/j.bej.2025.109708) and related PETases (Zheng 2024, 10.1021/acscatal.4c00400), which require extensive wet-lab screening yet explore only a narrow region

of sequence space due to their sequential, greedy nature. By leveraging the PROSS framework, which integrates multiple sequence- and structure-based filters, we systematically explored a much larger design space and identified several previously unreported stabilizing mutations and mutation combinations, including a set of four surface-charge-modifying substitutions that would likely not emerge from stepwise optimization. Finally, we strengthened the mechanistic interpretation by characterizing the structural and dynamic consequences of these mutations using X-ray crystallography and molecular dynamics simulations. These analyses provide a coherent molecular explanation for the observed improvements in stability and activity and help position our workflow as a generalizable and efficient route for engineering robust PET hydrolases.

2) Industrial benchmark conditions are inappropriate

The section “The R4 variants of PHL7 outperform ICCG and HotPETase in PET degradation under industrial process conditions” is problematic.

First, HotPETase should not be used as a benchmark, since multiple studies have already demonstrated that its performance under true industrial conditions is not superior (e.g. ACS Catal. 2023, 13, 13156–13166). More appropriate benchmarks would be ICCG-A2, TurboPETase, and Kubu-PM12, all of which have been tested under authentic industrial settings and shown to outperform ICCG. Second, the so-called “industrial conditions” employed in this study do not actually reflect standard industrial parameters. For example, the reactions for ICCG degradation (ref 12) were carried out in aqueous environments rather than in 0.1 M buffer solutions, and the substrate concentration was above 20% (w/w) PET powders, not the 10% (w/w) PET flakes used here. The use of PET flakes is particularly concerning. Some PETases exhibit higher degradation performance toward PET films compared to PET powders, likely due to differences in polymer surface adsorption and binding properties. However, all industrial protocols preferably use PET powders rather than PET films, as the increased surface area accelerates degradation and reduces processing time. Indeed, PET powders could be fully degraded within 24 hours by ICCG (for more details, please refer to ref 21 and ref 24 and ACS Catal. 2023, 13, 13156–13166). By contrast, the results presented in Figure 7C of this manuscript do not align with the previously reported industrial-level performance of ICCG. This discrepancy may be attributable to the use of PET films instead of powders, as well as the fact that enzyme concentrations may not have reached saturation. Therefore, I strongly recommend that the authors repeat these experiments under standardized industrial conditions—namely, using appropriate buffer systems, PET powder substrates at $\geq 20\%$ (w/w), sufficient enzyme loadings to reach saturation, and benchmarks that reflect state-of-the-art enzymes (A2, TurboPETase, Kubu-PM12). Without such standardized testing, the current dataset cannot provide reliable or comparable activity data for the field.

We have considered the concern regarding the choice of benchmarks and the need to evaluate enzyme performance under conditions that more closely reflect established industrial parameters. In response, we have expanded our benchmarking and experimental validation.

First, as recommended, we now include direct comparisons with state-of-the-art PET hydrolases. Specifically, TurboPETase and LCC-A2 were expressed, purified, and tested under the same reaction conditions used for the PHL7 R4 variants. Their PET hydrolysis activities have been incorporated into the revised Figure 7a and are discussed in the Results section. Briefly, at 70 °C, TurboPETase and LCC-A2 exhibited PET film degradation rates of 13.94 and 14.94 $\mu\text{m}/\text{h}$ in 0.1 M buffer and 15.99 and 23.11 $\mu\text{m}/\text{h}$ in 1.0 M buffer, respectively - values that fall within the range observed for the R4 variants (11.79-14.91 $\mu\text{m}/\text{h}$ in 0.1 M; 16.63-20.92 $\mu\text{m}/\text{h}$ in 1.0 M). At 65 °C, performance similarity was even more

pronounced. These additional experiments and comparisons, now integrated into the revised manuscript (Figure 7a and text on pages 20-21), address the reviewer's request for benchmarking against more relevant high-performance enzymes.

Second, to address the reviewer's concerns regarding industrial relevance, we conducted new bioreactor experiments using a higher substrate loading of 20 % (w/w) PET and included LCC-A2 as a top-tier industrial reference enzyme. Reactions were carried out in 0.1 M phosphate buffer, a condition under which both LCC-A2 and the R4 variants retain high activity and which has also been used in prior studies of LCC-A2 (Zheng 2024, 10.1021/acscatal.4c00400) and ICCG (Groseclose 2024, 10.1021/acscatal.4c04321). To rule out insufficient enzyme saturation, we performed each reaction at two enzyme loadings (0.8 mg/g PET and 0.4 mg/g PET). For these tests, we used the best-performing PHL7 variant, R4M10-H185Y, generated during paper revision by introducing the H185Y mutation - previously reported for FlashPETase (Wang 2025, 10.1016/j.bej.2025.109708) - to demonstrate the compatibility of our stabilized R4 scaffold with further engineering. The new results are illustrated in Figure 7e and described in the manuscript:

“The strong performance of R4M10 in the PET degradation bioreactor experiment indicated its suitability as a promising candidate for enzymatic PET recycling processes. To assess its capacity for further engineering and its robustness at elevated substrate loadings, we introduced one additional mutation, H185Y - recently reported to enhance the activity of WT PHL7 in FlashPETase - and evaluated the resulting R4M10-H185Y variant using EIS. The H185Y substitution increased the activity of R4M10 by 17-35%, demonstrating that the R4M10 scaffold can accommodate additional beneficial mutations. In 0.1 M buffer at 65 °C and 70 °C, R4M10-H185Y performed on par with the top-tier benchmark enzymes LCC-A2 and TurboPETase, and in 1.0 M buffer its activity was only marginally lower than that of TurboPETase (Figure 7a).

In a pH-stat bioreactor containing 20% (w/w) low-crystallinity PET at 68 °C in 0.1 M potassium phosphate buffer, R4M10-H185Y exhibited comparable performance to LCC-A2, which was selected as the benchmark due to its superior expression in *E. coli*. At an enzyme loading of 0.4 mg_{enzyme} g_{PET}⁻¹, R4M10-H185Y and LCC-A2 achieved 79% and 82% PET degradation, respectively, within 24 h (Figure 7e). Doubling the enzyme concentration to 0.8 mg_{enzyme} g_{PET}⁻¹ resulted in only modest increases (84% and 86% for R4M10-H185Y and LCC-A2, respectively), indicating that enzyme saturation had been reached.”

Regarding the use of PET flakes, we agree that PET powder provides a larger surface area and therefore permits faster degradation. However, we selected PET flakes because they are simple and inexpensive to prepare directly from post-consumer containers, require minimal mechanical preprocessing, and thus represent a practically relevant substrate for evaluating process feasibility. As there is currently no established industrial standard - given that PET enzymatic recycling has not yet advanced beyond pilot-scale operation - there remains room for alternative process concepts. The use of easily prepared flakes may offer advantages in terms of CO₂ footprint and economic viability by reducing energy-intensive pretreatment steps. Despite their larger particle size, we demonstrate that high degradation rates can still be achieved, reaching degradation rates of 100 g L⁻¹ within 10 h under our optimized conditions (see Figure 7e). We have clarified this rationale in the revised manuscript.

Taken together, we believe the expanded benchmarking, the new bioreactor experiments at higher substrate loadings, and the inclusion of industry-standard reference enzymes directly address the reviewer's concerns and provide a more rigorous and relevant assessment of PHL7 variant performance under conditions approaching industrial practice.

Modified Figure 7: PET film degradation by PHL7 R4 mutants and reference enzymes. **a** PET film degradation rates measured by EIS in $\mu\text{m per h}$ at 70 °C and 65 °C in 0.1 M or 1 M phosphate buffer. PHL7, its R4 variants, and R4M10-H185Y are compared to ICCG, HotPETase (HotP), LCC-A2, and TurboPETase (TurboP). Data are represented as mean \pm SD ($n = 3$). **b** Residual activity of the R4 variants, PHL7, and ICCG after incubation at 65 °C for 0.5, 1, 5 and 7 days. The activity was determined using PCL as substrate. Experiments were performed in triplicates. **c** Inverse Michaelis-Menten plots showing the rate of TPA release from PET determined after 4 h of reaction as a function of enzyme concentration for PHL7, R4M6, R4M9, and R4M10, respectively. Lines represent the best fit to the equation $v = \text{inv}V_{\text{max}} \cdot [E] / \text{inv}K_{\text{M}} + [E]$. The derived kinetic parameters are displayed in the inset table. **d** Low-crystalline PET flake degradation (10 % w/w) by R4 variants and ICCG at a concentration of 0.8 mg_{enzyme} g_{PET}⁻¹ in a pH-controlled bioreactor. The activity was calculated by determining the consumption of NaOH during the reaction (solid line) and by measuring the residual PET weight after stopping the reaction (point marker). Experiments were performed in duplicates ($n = 2$) for the R4 variants and in quintuplets ($n = 5$) for ICCG. The solid line represents the mean and the shaded area represents ± 1 SD. The inset graph shows the PET film degradation rate (in g_{PET}·L⁻¹·h⁻¹) of the R4 variants and ICCG. **e** Degradation of 20 % (w/w) PET by LCC-A2 and R4M10-H185Y in a pH-controlled bioreactor measured using NaOH consumption (solid line) or residual PET weight (point marker). Experiments were performed using an enzyme concentration of 0.4 mg_{enzyme} g_{PET}⁻¹ (left, $n = 2$) or 0.8 mg_{enzyme} g_{PET}⁻¹ (right, $n = 1$). The inset graphs show again the PET film degradation rate.

3) The manuscript emphasizes the impact of phosphate buffer concentration on enzyme stability, noting that PHL7 becomes unstable under 0.1 M phosphate (line 80). This suggests that the stability of these mutants may be significantly influenced by the solvent environment. However, the thermal stability (apparent melting temperature, T_m) of mutants is measured in a uniform buffer (50 mM Na₂HPO₄ pH 7.4, 200 mM NaCl) used for protein purification (line 863, line 807). This raises a critical concern: the T_m values obtained under these conditions may not reliably reflect stability in reaction environments, such as 1 M KH₂PO₄ buffer, or 0.1 M KH₂PO₄ buffer (without NaCl). The authors should directly compare T_m values of key variants (e.g., PHL7, and R4M6/R4M9/R4M10) in buffers mimicking actual reaction conditions. This would clarify whether the observed stability trends hold across different ionic strengths and salt compositions.

We agree that melting temperatures can depend on ionic strength. To address this, we measured the apparent T_m of PHL7 and of representative R4 variants (R4M6, R4M9, R4M10) in buffers reflecting the actual reaction conditions. As shown below in Supplementary Figure 16, T_m values obtained in the purification buffer (50 mM Na₂HPO₄, 200 mM NaCl) closely match those measured in 0.1 M KH₂PO₄, indicating that the stability data reported in the manuscript reliably represent behavior under low-salt reaction conditions, used e.g. in the bioreactor experiments. As expected, T_m values increased substantially, by ~11.2 °C for PHL7, when measured in 1.0 M KH₂PO₄. These comparisons confirm that the observed stability trends are consistent across the relevant buffer environments.

Supplementary Figure 16. Melting temperatures of selected WT PHL7 and selected R4 variants measured in different sodium phosphate buffer concentrations (0.05 M, 0.1 M, 1.0 M).

We have added the following text to the manuscript (page 20): “To directly assess the stabilizing effect of phosphate ions, apparent T_m values of PHL7 and selected R4 variants were determined across a range of phosphate buffer concentrations (Supplementary Figure 16). The results revealed a progressive increase in thermal stability, reaching T_m values above 95 °C for all variants at the highest concentration of 1.0 M.”

4) While the authors report >90% PET degradation for PHL7-R4 variants based on sodium hydroxide consumption in bioreactor experiments, this single metric may not fully capture the degradation process.

In contrast, reference 12 employs multiple complementary metrics for ICCG—including residual plastic dry weight, terephthalic acid (TPA) yield, and ethylene glycol (EG) yield—to robustly validate degradation efficiency. To strengthen the credibility of their findings, the authors should incorporate at least one additional quantitative measure (e.g., residual plastic dry weight or direct quantification of TPA/EG) alongside NaOH consumption.

PET dry weight data from the reactor experiments have been added, and Figure 7d+e has been updated accordingly (see above). The dry weight measurements (shown as points in Fig 7d+e) correlate well with the degradation levels calculated from NaOH consumption. Variants R4M6, R4M9, and R4M10 exhibited similar performance, achieving 98–99% PET degradation based on residual plastic weight, followed by R4M12 with 95.4% degradation. In comparison, ICCG degraded only 68.6% of PET on average. Although the dry weight–based values are slightly higher than those inferred from NaOH consumption, both methods show consistent trends and confirm that the PHL7 variants significantly outperform ICCG under the tested conditions (65 °C, 0.8 mg_{enzyme} g_{PET}⁻¹, 10% PET). Under the 20% (w/w) substrate loading conditions, the degradation levels calculated from NaOH consumption and residual PET weight also matched closely (Figure 7e).

5) Unclear presentation of computational pipeline

In the section "Rational design of PHL7 identifies activity enhancing mutations," the presented mutations are described as the final step of rational design rather than serving as starting points for subsequent computational iterations. This arrangement obscures the sequential logic of the engineering process, making it difficult for readers to track how each mutant evolves through successive optimization stages. The current depiction of mutant relationships in Figure 3a uses directional arrows that are prone to misinterpretation as iterative optimization. In reality, these variants were computationally generated and experimentally tested in parallel, without an experimentally validated stepwise optimization relationship. The multi-colored circular schematic in Figure 2a overwhelms readers with excessive visual complexity, hindering the extraction of key information. Overall, the narrative of computational design remains inaccessible to readers, undermining the technical credibility of the manuscript. The authors must restructure the workflow description and associated visuals to present a transparent, logically coherent engineering pipeline that aligns with standard conventions in computational enzyme design studies.

For a clearer and more logically structured presentation of the design workflow, we have created a new Supplementary Figure 3 that explicitly depicts the order in which mutations were introduced during the PROSS design process. Because PROSS generates designs along a mutation trajectory—where each successive variant inherits the mutations of the previous design and includes additional substitutions—we chose to represent this relationship using arrows between neighboring variants. To prevent misinterpretation of these arrows as indicating experimentally validated iterative optimization steps, we have added an explicit clarification to the caption of Figure 3a.

We would also like to clarify the narrative order in which the rationally engineered mutations (round 0) and the PROSS-generated variants (rounds 1 and 2) are presented. The manuscript begins with the R0 mutations because these were generated and tested first. However, as noted in the manuscript, this initial rational approach yielded only modest stability improvements, prompting us to pursue computational design using PROSS. The final design step then integrated beneficial mutations from both

approaches into the round 4 (R4) variants. We believe that this workflow is accurately summarized in Figure 1.

To further improve clarity and accessibility, we have moved Figure 1 to the beginning of the Results section and added a dedicated introductory paragraph summarizing the workflow depicted in this figure. We note that Reviewer 3 found this summary figure particularly helpful, and therefore we have retained its overall style while expanding the accompanying description to provide additional context.

In summary, while this manuscript presents a substantial dataset, including >60 engineered mutants, seven high-resolution structures, and MD simulations, its scientific contribution is limited by insufficient benchmarking against state-of-the-art enzymes, use of non-standard industrial conditions, and lack of theoretical advancement.

We appreciate the reviewer's constructive perspective on how the manuscript's impact could be strengthened. We believe that the additional benchmarking and control experiments we have now included - particularly the direct comparisons with state-of-the-art engineered PET hydrolases - substantially enhance the relevance of our findings. We also hope that our clarification of the PROSS-based design process, which explores sequence and structural space far more broadly than the sequential, greedy optimization schemes commonly applied in prior PETase engineering studies, sufficiently addresses the concerns regarding theoretical and methodological novelty. Finally, we believe that the combination of this design approach with our extensive new dataset, including a large set of high-resolution X-ray structures and complementary MD simulations, provides mechanistic insights that meaningfully advance current understanding of PETase stability and activity and will support future engineering efforts in this field.

Reviewer #2:

In this work, Blazquez-Sanchez and co-workers have computationally designed and synthesized a set of PHL7-based hydrolases for efficient PET degradation. After the strategy employed, the authors have proposed novel PHL7 variants with excellent properties and potential for industrial applications. The new variants (R4M6, R4M9 and R4M10) present with high thermal stability (88-95 °C) and can efficiently degrade PET in industrial conditions, even outperforming the reference LCC_{ICCG} enzyme under some conditions. Therefore, the outcomes of these article (the novel PHL7 variants) are highly interested and can contribute towards a green-technology for PET recycling. I recommend the publication of this study in *Nature Communications* but some points should be addressed to improve it.

1) The generation of the R2 variants is not totally clear and more detail would be helpful. R2M1 is based on R1M1 plus some mutations from R1M4 and R1M7 but not from other variants. R2M2 is based on R2M1 except two mutations (L176N and T219, why?) and again plus some mutations from R1M6 and R1M9 but not from other variants. This is not well explained. Probably, some graphs showing the evolution of energy as a function of the new mutations in R1M1-M9 variants from the Rosetta PROSS algorithm and a brief explanation in the Supporting Information can help understand the computationally proposed R2 variants.

We have added now a detailed description and graphical representation of the mutational design workflow used to generate the R1 and R2 variants to the Supporting Information.

“Supplementary description on the design of R1 and R2 variants: We designed nine PHL7 variants using the PROSS2 webserver (<https://pross.weizmann.ac.il/step/pross-terms/>). These first-round (“R1”) variants contained between 8 and 40 mutations (Supplementary Table 1) and were generated in a cumulative design scheme, where each successive variant incorporated all mutations from the preceding one along with additional substitutions. Based on computational evaluation, two second-round (“R2”) variants, R2M1 and R2M2, were constructed by manually combining mutations from selected R1 variants that exhibited improved Rosetta energy scores relative to both the wild-type (WT) enzyme and their respective predecessors in the design trajectory (Supplementary Figure 3). Specifically, R2M1 combined mutations from R1M1, R1M2, R1M4, and R1M7, because these variants showed an improved or equal score relative to their predecessors in the design trajectory. R2M2 incorporated the mutations of R2M1, excluding L176N and T219I (from R1M2), together with mutations from R1M6 and R1M9, which also showed favorable energy changes. Visual inspection of the structural models for R2M1 and R2M2 was performed to identify and exclude incompatible mutations. For instance, D233N from R1M7 was omitted to preserve a putative salt bridge between residues E148K and D233, and N213T from R1M9 was excluded to maintain a stabilizing hydrogen bond with S69. L176N and T219I were only tested in the R2M1 template but not in R2M2, because these mutations occur very close to the active site (e.g. L176 is next to the catalytic D177) and we expected losses in activity from these mutations.”

Lys148 - Asp233 salt bridge in R2M2 during MD

Lys148 - Asp233 interaction in HREX simulations

Supplementary Figure 6. K148-D233 interaction in MD simulations of R2M2 (top) and HREX simulations of R4 variants (bottom). Shown is the distance to the closest OD atom of D233, measured over 5*100 ns simulations.

3) Related to the previous comment, I would also extend the MD analysis for the changes seen for mutations N113D that induces a conformational change in S114 and the more water interactions due to V115T.

We have extended MD analysis and observed that N113D-triggered conformational changes, including Ser114 interaction profile, persist (Supplementary Figure 9). There was no overlap between conformations in WT and R2M2. In addition, we constructed RDF profiles for water molecules around the CB atom of residue 115, and observed that V115T substitution indeed resulted in an increased solvation (Supplementary Figure 10).

We have added these observations to the main text: “The backbone displacement associated with the N113D mutation in the R2M2 structure and the resulting alterations in the interaction profile of residue S114 were maintained throughout the R2M2 MD simulation (Supplementary figure 9). In addition, radial distribution function analysis of the water shell surrounding residue 115 confirmed that the V115T substitution leads to an increase in local solvation of the R2M2 protein (Supplementary figure 10).”

Supplementary Figure 9. Differences in S114 interactions between WT and R2M2 persist in MD simulations. Shown is data over 5 replicates 100 ns each.

Supplementary Figure 10. Radial distribution functions of water molecules oxygen atoms with respect to the position of CB atom of residue 115. Shown is data over 5 replicates 100 ns each.

4) The angles (φ , ψ) used to compare the flexibility of R2M2 with the WT PHL7 are not well defined.

The angles ϕ (phi) and ψ (psi) are standard Ramachandran angles used to characterize conformations of protein backbone. We added this clarification to the caption of the related Figure 5.

5) To try to understand better the low activity of R2M2, an extra structural/energetic analysis from the MD simulations can be done and focused on the catalytic triad. For instance, comparison between the hydrogen bonds between H209 and D177 or the electrostatic interaction between both units along the dynamics for R2M2 and WT PHL7 can help explain the low activity.

In the manuscript, we have reported triad-related comparisons for measurements that are indeed different between systems. The His209-Asp177 interaction geometry was not affected by any substitutions, in contrast to Ser131-His209 in substrate-free forms. We have added additional plots to the SI section (Supplementary Figure 11, 12, 13, 14) and an additional sentence to the main text to explain that:

“Moreover, the interaction geometry between H209 and the catalytic S131 in R2M2 exhibited a greater deviation from the optimal geometry (Supplementary figure 11, 12), whereas the configurational parameters of the D177-H209 interaction remained essentially unchanged (Supplementary figure 13, 14).”

Supplementary Figure 11. Influence of substitutions in R2M2 and R4 mutants on the interaction geometry of catalytic S131 and H209 in MD simulations of free enzyme forms. **a** Representation of assessed metrics - distances, angles, and torsions. **b** Histograms for free enzyme systems under 300K. **c** Histograms for free enzyme systems under 343K. Shown is data over 5 replicates 100 ns each.

Supplementary Figure 12. Influence of substitutions in R2M2 and R4 mutants on the interaction geometry of catalytic S131 and H209 in MD simulations of substrate-bound enzyme forms. a Representation of assessed metrics - distances, angles, and torsions. **b** Histograms for MHET-bound enzyme systems under 300K. **c** Histograms for MHET-bound enzyme systems under 343K. Shown is data over 5 replicates 100 ns each.

Supplementary Figure 13. Influence of substitutions in R2M2 and R4 mutants on the interaction geometry of catalytic H209 and D177 in MD simulations of free enzyme forms. a Representation of assessed metrics - distances, angles, and torsions. **b** Histograms for free enzyme systems under 300K. **c** Histograms for free enzyme systems under 343K. Shown is data over 5 replicates 100 ns each.

Supplementary Figure 14. Influence of substitutions in R2M2 and R4 mutants on the interaction geometry of catalytic H209 and D177 in MD simulations of substrate-bound enzyme forms. a Representation of assessed metrics - distances, angles, and torsions. **b** Histograms for MHET-bound

enzyme systems under 300K. **c** Histograms for MHET-bound enzyme systems under 343K. Shown is data over 5 replicates 100 ns each.

6) To rationalize the enhanced activity of the best R4 variants compared to R2M2 and WT PHL7, I would estimate from the MD simulations the substrate-enzyme binding energies to see if the recognition process is further facilitated for the R4 variants.

We have performed MMPBSA calculations for the results of HREX simulations of complexes of enzyme variants with 4xMHET substrate. We found that all R4 variants in fact have slightly lower affinity to this substrate than WT. It might mean that the increase in activity in R4 variants originated either from lower reaction barrier, or higher proportion of pre-reactive states, or both, but not from overall better binding. For WT, we analyzed MMPBSA scores for individual clusters based on the location of L1 ring, and found out that such a low score originates from the conformation unique (and prevalent) for WT. If this conformation is indeed associated with lower catalytic efficiency, this might explain the observed results. We should however note that these calculations mostly refer to interaction enthalpy. While the substrate and therefore its reorganization entropy is the same, the reorganization entropy of the binding site of the enzyme might differ between variants, influencing the actual interaction free energy that cannot be reliably estimated with this method.

We have added an extra paragraph to the man text (page 26): “MMPBSA analysis showed that the R4 variants displayed higher interaction energies than the WT (more positive values indicate lower binding strength) (Supplementary table 5), indicating that their enhanced catalytic activity is unlikely to result from increased substrate binding affinity. Rather, it may arise from a greater population of pre-reactive states, improved active-site organization, as discussed above, or other factors such as a reduced activation energy barrier. We further note that the entropic contribution to the binding free energy, associated with reorganization of the enzyme’s binding pocket and the substrate, cannot be reliably quantified using the MMPBSA approach and may differ substantially among variants.”

Supplementary Table 5. MMPBSA-computed interaction energies between enzyme variants and 4xMHET substrates in HREX trajectories. The trajectory from the replica with the unmodified topology is analyzed. Reported is the number of frames used for calculation, not the overall number of frames in trajectories.

	In oxyanion hole			Near oxyanion hole			Other poses		
	ΔE , kcal/mol	SEM	N frames	ΔE , kcal/mol	SEM	N frames	ΔE , kcal/mol	SEM	N frames
WT	-40.98	0.04	8576	-43.61	0.05	11077	-41.80	0.05	12501
R4M6	-37.01	0.06	9637	-43.61	0.04	15141	-42.93	0.05	12501
R4M9	-37.56	0.06	9166	-44.14	0.04	10726	-41.86	0.05	12501

R4M10	-37.72	0.06	8182	-45.17	0.05	11794	-44.04	0.05	12501
-------	--------	------	------	--------	------	-------	--------	------	-------

7) Additionally, more structural/energetic analysis of the catalytic triad in the R4 variants from the MD simulations in line with my comment above (5) can be useful to explain the enhanced activity.

The results of our MD analysis of the Ser131–His209 and Asp177–His209 interaction geometries in the R4 mutants, in both the substrate-free and MHET-bound states, are presented in our response to comment 5 (Supplementary Figures 11–14). Using an analogous approach, we examined the HREX simulation trajectories of the WT and R4 variants in complex with the 4xMHET substrate. The same trends were observed: the Ser131–His209 interaction exhibited substantial variant-dependent differences (Supplementary Figure 19), whereas the His209–Asp177 interaction remained largely unaffected by the substitutions (Supplementary Figure 20).

Supplementary Figure 19. Influence of substitutions in R4 variants on the interaction geometry of catalytic S131 and H209 in HREX simulations of 4xMHET-bound enzyme forms. **a** Representation of assessed metrics - distances, angles, and torsions. **b** Histograms for 4xMHET-bound enzyme systems.

Supplementary Figures 20. Influence of substitutions in R4 variants on the interaction geometry of catalytic H209 and D177 in HREX simulations of substrate-bound enzyme forms. a Representation of assessed metrics, distances and angles. **b** Data for 4xMHET-bound enzyme systems.

Reviewer #3:

This manuscript by Blazquez-Sanchez et al. presents a study on the engineering of a highly active derivative of a polyethylene terephthalate (PET) hydrolase that is thermostable in relatively low ionic strength. This hydrolase is being investigated for the industrial-scale degradation of plastics based on PET, but its dependence on high ionic strength for stability minimizes its use for industrial-scale bioreactor applications. Thus, the authors used a combination of *in silico* protein modelling, protein engineering, biochemical characterization and x-ray crystallography to generate an enzyme with the desired properties. The research represents a tour de force in its comprehensive design and performance, such that the underlying mechanisms driving the activity and thermostability could be discerned. This, and the importance of the research that addresses a major ecological concern involving plastic waste pollution, impacts the significance of the work.

This reviewer has only two issues the reviewers may want to address. The first concerns the last paragraph of the Introduction which, in effect, is a reiteration of the Abstract. This could be replaced with a simple statement of purpose, and Figure 1 could be integrated into the Results section, perhaps involving an introductory statement regarding experimental design.

Following the reviewer's suggestion, we have replaced the last paragraph in Introduction by a statement of the study aims and have moved Figure 1 together with a description on experimental design to Results.

"Accordingly, this study aims to overcome current limitations of PHL7 by engineering variants with enhanced thermostability, catalytic efficiency, and reduced salt dependence, thereby advancing their suitability for scalable, industrial PET recycling processes."

"PHL7 was engineered through an iterative design workflow combining Rosetta-based protein modeling with rational mutagenesis to systematically enhance enzyme stability and activity (Figure 1). Efficient exploration of a broad sequence and structural space by the PROSS algorithm³¹ enabled the identification of beneficial multi-mutation combinations that would likely remain undiscovered in conventional single-site mutagenesis strategies. Successive design-test cycles across four engineering rounds, described below, identified mutations that optimized the balance between thermostability and catalytic efficiency. This approach established a robust framework for the rational improvement of PET hydrolases."

The second, and also a relatively minor issue, pertains to the experimental design of the thermostability analysis (Melting Point Determination). As a major focus of this research was to generate a hydrolase with enhanced stability in low ionic strength (0.1 M phosphate buffer), compared to its current property of requiring 1 M concentrations, it is curious that the thermostability assays were conducted in neither conditions, but rather something more in between (*viz.* 50 mM phosphate with 200 mM NaCl).

The apparent melting temperature depends indeed on the used phosphate buffer concentration, with higher ionic strength leading to increased stability. To demonstrate this, we have measured T_m values of PHL7 and representative R4 variants (R4M6, R4M9, R4M10). The results are shown in Supplementary figure 16 and mentioned in the text.

Supplementary Figure 16. Melting temperatures of selected WT PHL7 and selected R4 variants measured in different sodium phosphate buffer concentrations (0.05 M, 0.1 M, 1.0 M).

“To directly assess the stabilizing effect of phosphate ions, apparent T_m values of PHL7 and selected R4 variants were determined across a range of phosphate buffer concentrations (Supplementary Figure 16). The results revealed a progressive increase in thermal stability, reaching T_m values above 95 °C for all variants at the highest concentration of 1.0 M.”

Reviewer #4:

This work follows on from the discovery and characterization of a thermotolerant PETase, PHL7, that was isolated from a compost metagenome. While this enzyme showed promise in terms of its stability and activity profiles relative to some benchmark enzymes currently used for PET deconstruction, this particular enzyme required unusually high salt concentrations, up to 1 M phosphate, to achieve this. As noted by the authors, this represents a significant limitation for the use of this enzyme in an industrial setting where buffer and salt requirements should be minimized, as demonstrated in multiple TEA and LCA studies. However, the advantage of enzymes that can maintain high activity at elevated temperatures, ~65 deg C, and even remain active over multiple cycles, remains a key target for the field.

In this study, the native enzyme is engineered to produce a performant variant that retains/surpasses the activity profile, increases the thermotolerance, and mitigates the requirement for high salt conditions. In fact, and a rather a hidden outcome of this work up front, is that one of their variants retained full activity after a week at 65 deg C (see comment below). This is a well conducted study that leverages significant expertise in the team in terms of protein engineering and analysis. The combination of modelling, X-ray crystallography, and stability/activity measurements is well executed and presented.

1) While most PETase enzymes require some salt to maintain their optimum activity, PLH7 is rather an outlier in terms of the extreme requirement for molar concentrations. In the introduction, high salt requirement is framed as a general issue. To be more transparent to the reader, it would be better to note that several benchmark enzymes (as described), have more modest salt requirements. Published Kubu-P bioreactor experiments were performed at 100 mM Na₂HPO₄ for example. Thus, the focus here is on reducing the unusually high salt reliance specifically for PLH7 in order to bring it closer in alignment to the industrially relevant benchmarks, including LCC variants and Kubu-P, which is indeed the target and successful outcome here. This remains a valid goal, but rewording will helpfully delineate this study from engineering this parameter in PETases more generally.

We agree that lowering the high salt concentration dependence has been a design goal especially for PHL7. To make this point clearer, we have rephrased the second paragraph in introduction and added the following sentence:

“While some PET hydrolases already perform well under low-salt conditions, others depend on elevated ionic strength to remain stable and catalytically active, rendering the mitigation of salt dependence a design goal for specific enzymes.”

2) The strategy is well considered and nicely depicted in Figure 1, making a complex and involved process generally accessible. However, I suggest the each of the sections that describe the 4 rounds are shortened for further readability. Some of this text could be productively moved to the SI, allowing the main text to focus on more concise strategies and outcome for each. Listing every mutation and result in the text is distracting from the main messages, particularly when much of these data are shown in the figures. Examples, lines 422-435, lines 473-509, where the text is rather dense.

Following this suggestion, we have rephrased the manuscript sections on the engineering of R3 and R4 variants to make them more concise. The paragraph between lines 422-435 reads now:

“To evaluate potential synergistic effects, the most impactful single mutations – P155G, A186S, and D208S – were combined into double and triple variants. As shown in Figure 6a, variants containing P155G exhibited reduced thermostability relative to R2M2, whereas the A186S/D208S combination

(R3M8) retained high thermal stability. Notably, the triple variant R2M2-P155G/A186S/D208S (R3M9) demonstrated the highest catalytic activity (1.04-fold), comparable to that of PHL7 in 1.0 M phosphate buffer and at 70 °C.”

And the paragraph between lines 473-509 was shortened to:

“All variants derived from R3M4 retained high thermostability (>95 °C; Figure 6a, Supplementary Figure 2). Among these, progressive combinations of L104Q, Q175E, D208S, and L210T led to the most pronounced enhancement in activity, with variant R4M8 reaching wild-type levels (1.01-fold). Additional modifications of R4M8, including variant R4M11, which carried a salt bridge-swapping double mutation D233K/K148E, and variants R4M13 (L93F/Q95G) and R4M14 (L93F/Q95Y), failed to yield further improvements.

In the R3M9 background, stepwise introduction of L104Q, Q175E, and L210T produced variants with progressively higher stability and activity, culminating in R4M6, which exhibited the strongest enhancement relative to the WT (1.33-fold activity). Subsequent introduction of the salt bridge-swapping mutation D233K/K148E produced variant R4M12, which retained a similar T_m and activity (1.37-fold). Positions 93 and 95 were also investigated in the R4M6 template by introducing the mutations L93F/Q95G and L93F/Q95Y, resulting in variants R4M9 and R4M10, respectively. R4M9 showed a substantially increased T_m (>95 °C), whereas R4M10 exhibited modest T_m and activity increases compared to R4M6. Overall, R4M6, R4M10, and R4M12 emerged as the top-performing variants from this round, providing the most favorable trade-off between catalytic efficiency and thermostability (Figure 6a).”

3) It would be interesting to measure the actual T_m of variant R2M2 with a compatible method (line 240), such as DSC. Without these data, this should not be plotted as a single data point in Fig 3a. Similarly, the delta T_m for this variant is unknown and may be higher (Fig 3b).

To clarify that the T_m of R2M2 and other R3 and R4 variants is higher than 95°C, we have added an extra sentence to the caption of Figure 3 and Figure 6.

“Note that the T_m of R2M2 is >95 °C.”

“Note that the T_m of variants marked with an asterisk is >95 °C”

4) The significance of the stability of the R4M9 variant is somewhat understated. Continuous processing and/or enzyme recovery are limited by long-term stability, but have great potential to positively transform the recycling process. While out of the scope experimentally of this study, it would be interesting to see the longevity limits here, towards the goal of months for example. A comment in the conclusions would be welcome, including the potential translation of these properties to other enzymes with similar scaffolds.

We have added an extra sentence to Discussion (page 29) to emphasize the considerable long-term stability of R4M9 and how a translation of this property to other enzymes could help in the development of continuous enzymatic recycling processes.

“The exceptional durability of R4M9 highlights its potential as a template for designing PET hydrolases suitable for continuous recycling processes with enzyme recovery.”

5) Please report the crystallinity of all the substrates used in the main text: the semi-crystalline PET films (line 531), the low-crystalline PET flake (line 584), and low-crystalline clamshell (line 901). In the

industrial context that is presented, a comment should be made on the performance of these enzymes on model substrates with defined high crystallinity, as compared to the amorphous films that was tested.

We have now specified the crystallinity of the used PET films and PET containers, which were determined by DSC in our previous paper (Sonnendecker et al. (2022) ChemSusChem 15).

“PET films of 3 cm x 0.5 cm (about 45 mg) (3-5 % crystallinity) (Goodfellow, Hamburg, Germany)”
“The crystallinity of these PET containers was previously determined to be between 3-5 %²⁶.”

We have added a comment on the limitations of PHL7 in degrading high-crystalline PET to the Discussion (page 29+30):

“Another key determinant of the industrial applicability of PET hydrolases is their activity on highly crystalline PET, as the majority of PET waste – particularly PET bottles – exhibits crystallinity levels above 25 %. Previous studies have demonstrated that the catalytic performance of PHL7 and other hydrolases declines sharply when substrate crystallinity exceeds approximately 20 %⁴⁹. Accordingly, a critical future objective is the development of PHL7 variants with improved tolerance to high crystallinity, facilitating more efficient degradation of industrially relevant PET materials.”

6) The reverse experiment for the charged mutations is well considered and insightful. An SI figure (and main text commentary) plotting the pI of the variants, and including other known proteins, would be helpful for context. (around lines 275-286)

We have calculated theoretical pI values (with ExPASy ProtParam tool) for PHL7, some engineered PHL7 variants and some reference PET hydrolases used in this study. The results of this analysis are shown in Supplementary Figure 4 and mentioned in the text (page 10):

“Consistent with this, PHL7 has a low isoelectric point (pI) of 5.5, which increases to 6.2 in R2M2. In contrast, other PET hydrolases used for comparison in this study (e.g. ICCG) exhibit substantially higher pI values (8.5–9.5), reflecting their lower number of negatively charged residues (Supplementary figure 4).”

Supplementary Figure 4. Predicted isoelectric points (pI) of PHL7, PHL7 variants and other PET hydrolases. The pI values were calculated from the protein sequences using the ExPASy webserver (<https://web.expasy.org/protparam/>).

7) The lack of major structural rearrangements, coupled with the subtle changes observed by MD are intriguing, given their significant biophysical effects on activity. While the structural comparisons (including in the SI) are comprehensive between the parents and engineered variants, it would be valuable to have a comparison between at least the wt and an R2 or R4 variant with LCC_ICCG or similar as a better segway to the activity data that follows. (or alternative in section beginning line 608). It would be interesting to see if there is a trajectory towards higher similarity with some of the benchmark enzymes over the rounds.

We agree that understanding structural and dynamic differences among enzyme variants is essential for interpreting their divergent activities. To this end, we have carried out extensive molecular dynamics simulations for key variants - including R2M2, R4M6, R4M9, and R4M10 - in both substrate-free and substrate-bound states (with MHET and 4×MHET). We analyzed active-site geometry, hydrogen-bonding patterns, and the relative populations of bound and pre-reactive conformations, as these features are directly linked to catalytic performance. The principal results are presented in Figures 5 and 8 of the main text, and additional analyses have been incorporated into the revised manuscript as Supplementary Figures 11–14 (see our responses to Reviewer 2).

Across these simulations, we consistently observed more favorable active-site pre-organization in the highly active R4 variants and in WT PHL7 compared to the less active R2M2. Moreover, the HREX simulations with the 4×MHET substrate revealed higher proportions of pre-reactive states in the R4 variants, supporting the conclusion that their improved activities arise from more efficient formation of catalytically competent binding geometries.

Regarding the suggestion to compare these structures or MD trajectories with ICCG or related benchmark enzymes: the active site architecture and substrate-binding pocket of ICCG differ substantially from those of PHL7 and its engineered derivatives, involving multiple non-conservative amino-acid substitutions. Given these divergent structural frameworks, we do not expect a progression toward increased similarity to ICCG as PHL7 variants become more active. For this reason, and in order to maintain focus on mechanistically informative comparisons within the PHL7 evolutionary trajectory, we have not performed additional MD simulations for ICCG.

8) In the conclusions, it would be interesting to reflect on the reasons behind the native enzyme's high phosphate requirements, and link to the observation that this may correlate with the acid patch on the wt enzyme.

We agree with the reviewer's assessment and hypothesize that PHL7's requirement for high potassium phosphate concentrations arises from its highly negative surface charge, which necessitates elevated ionic strength for charge compensation. This interpretation is supported by our charge-reversion experiments described above. We have clarified this point in the Discussion section as follows:

“Our results are in good agreement with previous observations on halophilic proteins^{37,47,48}. Like PHL7, they are coated on their surfaces with a high number of Asp and Glu residues. Halophilic proteins require high salt concentrations to compensate their high charge density and prevent protein unfolding, which may occur at low salt concentrations due to electrostatic repulsions.”

9) The commentary around the mechanistic model for PET surface binding and interfacial catalysis could be more easily described with reference to an SI figure depicting the likely position of the PET surface in relation to the modified charged amino acids.

We have added a reference to a schematic of the proposed surface-binding model presented previously by Thomsen et al. (ChemSusChem, 2023; <https://doi.org/10.1002/cssc.202300291>, Figure 7). This figure

illustrates the position of the negatively charged surface patch - corresponding to the region targeted by our charge-neutralizing mutations - on PHL7 in relation to the PET surface. The revised text (page 28) now reads:

“According to the proposed mechanistic model illustrated by Thomsen et al.⁴⁹,

Reviewer #1:

The revised manuscript has addressed most concerns from the first review, particularly by improving experimental validation, data presentation, and benchmarking. The added comparisons against recently reported enzymes (e.g., TurboPETase, LCC-A2) and the use of multiple degradation metrics (NaOH consumption and residual PET weight) improve transparency and reproducibility. These revisions strengthen the experimental reliability and readability. However, critical limitations remain regarding (i) methodological innovation and (ii) marginal industrial-scale benefit relative to frontier bottlenecks. In its current form, the study is best described as a competent application of established protein design tools to a single enzyme scaffold, with incremental mechanistic rationalizations that do not yield a new engineering principle generalizable beyond the presented case.

The authors frame the work as an “in silico–first, multipoint design methodology” that “substantially reduces experimental testing effort” and argue that the PROSS-based process explores sequence space more broadly than “sequential, greedy optimization schemes”. Yet, the computational component relies on existing, widely used infrastructure (Rosetta/PROSS) and is even executed via the PROSS2 webserver. I did NOT find (a) a new scoring function, (b) a new predictive model/learning strategy, or (c) a new sampling/search algorithm. The field already contains numerous examples where existing computational stabilization/design frameworks (e.g., PROSS/FireProt/FRESCO) are used to reduce experimental burden and enable multipoint mutagenesis—this is an established paradigm rather than a novelty claim. Recently, several high-impact PETase papers that are “method-driven” make their novelty very explicit—e.g., the VenusMine pipeline integrates language-model embeddings + representation-tree clustering with structure/sequence retrieval and multi-tier screening. process integrates language model embedding and representation tree clustering, combined with structure/sequence retrieval and multi-layer screening (Nat Commun 16, 6211 (2025)); a study published in Science in 2025 developed a "landscape construction and sampling" framework, utilizing neighborhood analysis and hierarchical/cluster sampling to track the fitness landscape (Science 387, eadp5637(2025)). Against such standards, the present manuscript’s computational component does NOT read as a methodological advance. Given this context, the authors should substantially tone down statements implying algorithmic or methodological innovation, and instead clearly position the work as an application/engineering case study.

And more importantly, even after revision, the engineered variants appear to match—but not convincingly exceed—the performance envelope already achieved by leading PET hydrolases reported in the last 1–2 years (e.g., TurboPETase and LCC-A2) under standardized benchmarking. In this context, the manuscript reads primarily as a catch-up study: it arrives ~2 years later at a similar performance level, yet does not deliver either a new design paradigm or a demonstrable process-level advantage. The study still does not engage sufficiently with the frontier bottlenecks emphasized in current PET biorecycling research: degradation of high-crystallinity PET, performance under realistic mixed-waste streams, and process concepts that reduce alkali dependency / downstream neutralization burden (Nat. Chem. Eng. 2025, 2, 309–320). As a result, the contribution appears incremental and non-leading in time, raising a

fundamental question of editorial fit: if comparable performance was demonstrated ~2 years ago, what is the distinct, current advance that warrants publication at the same journal tier now? In the absence of a new methodology, new mechanistic principle, or clear process-level breakthrough, I do not see a compelling justification.

Overall, in its current form, the work does not provide a new computational paradigm nor a step-change in PET biorecycling performance under frontier-relevant constraints, and thus I do not yet see a strong justification for Nature Communications.

We thank the reviewer for their careful second-round evaluation and for acknowledging that the revised manuscript substantially improves experimental validation, benchmarking, and transparency. We would like to further clarify the scope, positioning, and distinct contribution of our work.

We agree that this study does not aim to introduce a new computational design approach comparable to recent method-driven papers. We conceptualized our work as a rigorously executed, in silico-guided engineering case study applied to a challenging and underexplored PETase scaffold, rather than as a methodological advance in computational enzyme design. Accordingly, we have framed the computational design of PHL7 variants as an application case and omitted any claims of algorithmic or methodological novelty in the revised manuscript. Importantly, however, we emphasize that the contribution of this study lies not in computational novelty per se, but in (i) the choice of design target, (ii) the operational constraints addressed, and (iii) the mechanistic insights derived from extensive structural and biophysical characterization.

With respect to the design target, we respectfully disagree that this work constitutes a simple “catch-up” study. PHL7 represents a distinct polyesterase lineage and is not interchangeable with LCC-derived enzymes such as LCC-A2 or TurboPETase. Prior to this study, PHL7 variants were constrained by narrow operating windows and a strong dependence on high buffer concentrations, and structure–stability–activity relationships for this scaffold were only sparsely explored. Expanding the portfolio of high-performance PETases beyond a small number of closely related scaffolds is important for industrial translation, where robustness, expression yield, folding behavior, and process compatibility can differ substantially even among enzymes with similar apparent activities.

Crucially, the advance reported here lies in the operational regime achieved rather than in peak activity values alone. The engineered PHL7 variants operate at elevated temperatures under reduced buffer concentrations, directly addressing alkali consumption and downstream neutralization burden, which the reviewer correctly identifies as a frontier bottleneck. In addition, our work demonstrates that high thermostability in PETases can be achieved without reliance on artificial disulfide bonds, which have become a common but often suboptimal stabilization strategy. In contrast to enzymes such as ICCG, TurboPETase, or Kubu-PM12, the PHL7 variants reported here do not require engineered disulfide bonds to maintain activity at

high temperatures. Avoiding disulfide engineering can reduce risks associated with misfolding, redox sensitivity, and reduced expression yields, thereby offering a tangible advantage for large-scale enzyme production and process economics.

Beyond empirical optimization, this study provides mechanistic insight into PHL7 stabilization and activation. We identify and structurally rationalize a coherent set of stabilizing mutations distributed across surface charge modulation, core packing, and loop rigidification, supported by X-ray crystallography, molecular dynamics simulations, and experimental validation. These analyses move beyond post hoc rationalization and contribute transferable design insights, including negative design lessons such as when disulfide-based stabilization may be undesirable.

In summary, we believe that Nature Communications remains an appropriate venue for this work because it delivers a deeply characterized PETase variant operating under previously limiting, process-relevant constraints; diversifies the set of viable high-performance PET hydrolase scaffolds; and provides mechanistic clarity that informs future enzyme engineering beyond this specific case. The study complements recent method-driven advances by demonstrating how established computational tools, when rigorously applied and coupled to extensive mechanistic analysis, can yield meaningful and industrially relevant progress.

Reviewer #2:

The authors have reasonably addressed almost all comments raised by the reviewers and, the current, version has improved with respect to the previous one. In this regard, I recommend the publication of this study in Nature Communications.

Nevertheless, I think that the generation of the R2 variants is still confusing. For instance, I do not understand why the authors have incorporated some mutations from R1M7 to generate R2M1 and R2M2 according to an energetic criterion. From Supplementary Figure 3, I would have selected mutations from R1M1, R1M2 and R1M4 to propose R2M1 and mutations from R1M6 and R1M9 plus R2M1 to generate R2M2. The clarification of this point can be relevant for others groups which can adopt this computational strategy for other enzyme engineering applications.

We thank the reviewer for this thoughtful comment and agree that further clarification of the R2 variant construction is valuable for readers interested in adopting this design strategy. While Rosetta total energy served as an important guiding metric, it was not used as the sole criterion for selecting mutations for R2M1 and R2M2. As outlined in the Supplementary Methods describing the R1 and R2 designs, individual mutations were additionally evaluated for their structural and energetic compatibility with other substitutions. Mutations predicted to preserve or introduce favorable local interactions were retained, whereas those expected to cause unfavorable or clashing interactions were excluded.

Based on this combined assessment, we incorporated the mutations N113D and D196S from R1M7 into R2M1, while excluding D233N. The N113D substitution was expected to introduce a stabilizing hydrogen-bond interaction with the neighboring H109Y mutation, which had been introduced via R1M4. Specifically, the COO group of D113 is compatible with hydrogen bonding to the OH group of Y109, whereas the NH₂ group of the native N113 would be less favorable in this context. The D196S mutation was retained due to its contribution to a more neutral surface charge distribution. In contrast, D233N was excluded from both R2M1 and R2M2 to preserve a putative salt bridge between the mutation E148K and the native D233 residue, as described previously in the Supplementary Methods.

We have expanded the Supplementary Methods to clarify this rationale and to emphasize that the R2 variants were generated through an informed combination of Rosetta energy evaluation and structural compatibility analysis based on visual inspection of the models.

“Importantly, in addition to total Rosetta energy, individual mutations were also evaluated for their structural and energetic compatibility with previously introduced substitutions.”

“From R1M7, the mutations N113D and D196S were retained based on their predicted local stabilizing effects and compatibility with mutations introduced earlier in the design trajectory. In particular, N113D was expected to form a favorable hydrogen-bond interaction with the neighboring H109Y mutation (introduced via R1M4), while D196S was selected due to its contribution to a more neutral surface charge distribution.”

In this work, Blazquez-Sanchez and co-workers have computationally designed and synthesized a set of PHL7-based hydrolases for efficient PET degradation. After the strategy employed, the authors have proposed novel PHL7 variants with excellent properties and potential for industrial applications. The new variants (R4M6, R4M9 and R4M10) present with high thermal stability (88-95 °C) and can efficiently degrade PET in industrial conditions, even outperforming the reference LCC^{LCCG} enzyme under some conditions. Therefore, the outcomes of these article (the novel PHL7 variants) are highly interested and can contribute towards a green-technology for PET recycling. I recommend the publication of this study in *Nature Communications* but some points should be addressed to improve it.

- 1) The generation of the R2 variants is not totally clear and more detail would be helpful. R2M1 is based on R1M1 plus some mutations from R1M4 and R1M7 but not from other variants. R2M2 is based on R2M1 except two mutations (L176N and T219, why?) and again plus some mutations from R1M6 and R1M9 but not from other variants. This is not well explained. Probably, some graphs showing the evolution of energy as a function of the new mutations in R1M1-M9 variants from the Rosetta PROSS algorithm and a brief explanation in the Supporting Information can help understand the computationally-proposed R2 variants.
- 2) From the crystal analysis of R2M2, the authors point out the effects of some mutations. For instance, they mention the possibility of a salt bridge between K148 and D233. This bridge might not continue to be formed in the reaction conditions. This can be easily analysed from the MD simulations. Note that in a recent paper (J. Am. Chem. Soc. 2023, 145, 19243) it was shown that the salt bridge between E204 and K233 (originally supposed to be a stabilizing mutation) does not persists along a MD simulation and had a different effect.
- 3) Related to the previous comment, I would also extend the MD analysis for the changes seen for mutations N113D that induces a conformation change in S114 and the more water interactions due to V115T.
- 4) The angles (ϕ , ψ) used to compare the flexibility of R2M2 with the WT PHL7 are not well defined.
- 5) To try to understand better the low activity of R2M2, an extra structural/energetic analysis from the MD simulations can be done and focused on the catalytic triad. For instance, comparison between the hydrogen bonds between H209 and D177 or the electrostatic interaction between both units along the dynamics for R2M2 and WT PHL7 can help explain the low activity.
- 6) To rationalize the enhanced activity of the best R4 variants compared to R2M2 and WT PHL7, I would estimate from the MD simulations the substrate-enzyme binding energies to see if the recognition process is further facilitated for the R4 variants.
- 7) Additionally, more structural/energetic analysis of the catalytic triad in the R4 variants from the MD simulations in line with my comment above (5) can be useful to explain the enhanced activity.